# Parameter Decorrelation via Transition-Variance Alignment for Multivariate Time-series Forecasting

Ji-Eun Choi [* 1]   Jae-Hong Lee [* 2]   Joon-Hyuk Chang [3]

## Abstract

Multivariate time-series forecasting (MTSF) learns from high-dimensional covariates with strong temporal dependence, periodic structure, and cross-variable correlations. While modern pipelines often mitigate non-stationarity through instance-wise normalization and decomposition, these interventions operate at the data level and do not directly control dependence that can emerge among the parameters during training. We study MTSF optimization from a parameter-decorrelation viewpoint. Modeling stochastic optimization as a Markov chain in parameter space and leveraging its stochastic differential equation interpretation, we use the per-step transition-variance induced by gradient noise as a tractable signal for optimization-induced dependence and update uncertainty. This signal can empirically inflate during training; we theoretically show that such inflation can degrade generalization diagnostics. Motivated by this mechanism, we propose transition-variance alignment (TVA), an architecture-agnostic procedure that regulates transition-variance by smoothly gating the step size based on the mismatch between an estimated noise scale and a chosen target. TVA maintains effective transition-variance near a prescribed scale without architectural changes, incurs negligible overhead, and integrates seamlessly with diverse methods. Across real-world multivariate benchmarks, TVA consistently improves forecasting accuracy.

---

[*]Equal contribution  [1]Department of Artificial Intelligence, Hanyang University, Seoul, Republic of Korea [2]Division of Language & AI, Hankuk University of Foreign Studies, Seoul, Republic of Korea [3]Department of Electronic Engineering, Hanyang University, Seoul, Republic of Korea. Correspondence to: Joon-Hyuk Chang <jchang@hanyang.ac.kr>.

*Proceedings of the 43 rd International Conference on Machine Learning*, Seoul, South Korea. PMLR 306, 2026. Copyright 2026 by the author(s).

## 1. Introduction

Time-series forecasting (TSF) aims to predict future values from historical observations and supports planning and risk mitigation in domains such as energy systems, traffic management, and weather prediction (Quinlan, 1986; Cortes & Vapnik, 1995; Vapnik et al., 1996; Friedman, 2001; Barlin et al., 2013; Chen & Guestrin, 2016). In multivariate time-series forecasting (MTSF), the forecaster additionally exploits covariate signals to improve predictive performance, yielding a practically important but statistically more demanding setting (Wen et al., 2023; Qiu et al., 2024; Luo & Wang, 2024). The central challenge is to learn temporal dynamics in a high-dimensional covariate space that is both cross-correlated and strongly temporally dependent.

Deep neural networks (DNNs) have become a dominant solution paradigm for MTSF, spanning architectures from linear and multilayer perceptron (MLP) models to Transformer-based forecasters for long-horizon prediction (Zeng et al., 2023; Zhang & Yan, 2023; Nie et al., 2023; Lin et al., 2024a; Cai et al., 2024; Liu et al., 2024). Despite strong empirical progress, time-series data commonly exhibit pronounced periodicity and inter-sample dependence; consequently, training examples obtained from sliding windows are far from independent and identically distributed (i.i.d.) (Lin et al., 2024a). This mismatch can distort optimization dynamics and exacerbate generalization issues, particularly in long-horizon settings where repetitive structure and non-stationary shifts coexist.

A substantial body of work addresses these challenges via data decorrelation. Instance-wise normalization methods (Kim et al., 2022; Fan et al., 2023; Liu et al., 2023; Ye et al., 2024) normalize each sample using its own statistics, attenuating nuisance components such as sample-specific bias and scale before restoring them at the output. Complementarily, decomposition-based models (Zeng et al., 2023; Yu et al., 2024; Lin et al., 2024a) separate trend/seasonality or periodic components from residuals to isolate predictable structure. These methods operate primarily at the level of representation and have proven effective for mitigating distribution shift and periodic confounds. However, they do not explicitly regulate correlations that may emerge among parameters when the model is repeatedly updated on tempo-

rally dependent data.

In this work, we highlight the risk of over-correlated parameters in MTSF training and propose a mechanism to diagnose and mitigate it. We view stochastic optimization as inducing a discrete-time Markov chain on parameters and approximate its dynamics via stochastic differential equations (SDEs) (Li et al., 2019; 2021; Malladi et al., 2022). Under this perspective, each training step defines a transition kernel whose variance captures the uncertainty injected by gradient noise at that step. We use this transition-variance as a practical signal of optimization-induced dependence and update uncertainty. Empirically, we observe that the signal can inflate rather than stabilize; analytically, we show that such inflation can enlarge PAC-Bayes (Seeger, 2002) complexity terms, providing a mechanistic diagnostic for reduced generalization.

Guided by this insight, we propose transition-variance alignment (TVA), a simple architecture-agnostic method for controlling transition-variance during training. At each iteration, TVA estimates the transition-variance for the gradient-noise scale and compares it to a target level. TVA then smoothly gates the step size using a log-ratio discrepancy rule, shrinking the effective step size when the estimated noise scale deviates from the target and leaving it near its baseline when the scales are well matched. This adaptive regulation keeps the effective transition-variance close to the target, preventing runaway growth of the signal and stabilizing the induced parameter distribution. TVA requires no changes to the forecasting backbone and introduces only lightweight computations, making it compatible with linear, MLP, and Transformer-based forecasters. We validate TVA across diverse MTSF backbones, showing consistent improvements in training stability and forecasting accuracy on real-world benchmarks.

## 2. Related Works

### 2.1. Time-series Forecasting Models

TSF has long been studied through statistical modeling and has increasingly incorporated machine learning techniques to capture nonlinear temporal dependencies and complex covariate effects (Quinlan, 1986; Cortes & Vapnik, 1995; Vapnik et al., 1996; Friedman, 2001; Barlin et al., 2013; Chen & Guestrin, 2016). As deployments have grown in scope and dimensionality, research emphasis has shifted toward MTSF, which leverages multiple covariates to improve predictive accuracy and robustness (Wen et al., 2023; Qiu et al., 2024; Luo & Wang, 2024). Contemporary MTSF methods span a broad architectural spectrum, ranging from linear and MLP-based models to Transformer-style forecasters designed for long-horizon prediction (Zeng et al., 2023; Nie et al., 2023; Zhang & Yan, 2023; Lin et al., 2024a;

Cai et al., 2024; Liu et al., 2024). A recurring theme is the trade-off between architectural complexity and inductive bias, with several works reporting that carefully designed lightweight models can be highly competitive, while others show that attention-based mechanisms can be effective when paired with suitable tokenization schemes, channel modeling choices, or structural priors (Nie et al., 2023; Das et al., 2023; Lin et al., 2024a). Our work highlights a complementary dimension by showing how temporally dependent sampling, and the resulting optimization dynamics, shape the learned parameter distribution.

### 2.2. Data Decorrelation

A prominent strategy for improving forecasting under non-stationarity is to reduce redundant or unstable components in the input through normalization. Instance-wise normalization methods compute sample-specific statistics and normalize each window independently, suppressing nuisance factors such as shifting level and scale (Ogasawara et al., 2010; Passalis et al., 2019; Kim et al., 2022; Fan et al., 2023; Liu et al., 2023). Reversible normalization frameworks introduce a normalize–forecast–denormalize pipeline that removes window statistics prior to prediction and reinserts them at reconstruction time, improving robustness to distribution shift (Kim et al., 2022). More recent variants operate at a finer temporal granularity to better accommodate within-window non-stationarity (Fan et al., 2023; Liu et al., 2023). Frequency-aware normalization has also been proposed to address evolving seasonal components by operating in the frequency domain (Ye et al., 2024). Another widely used family of approaches performs seasonal–trend decomposition (STD), decomposing each input window into components such as trend and seasonality that are modeled separately (Wu et al., 2021; Zhou et al., 2022; Fan et al., 2022; Zeng et al., 2023). Many STD-based designs use moving-average filters to extract slowly varying trends, while more flexible approaches replace fixed filters with learnable decomposition kernels to adapt to dataset-specific dynamics (Yu et al., 2024; Lin et al., 2024b). By isolating periodic structure from residual variation, decomposition can stabilize learning and improve extrapolation. Normalization and decomposition primarily target data correlation and non-stationarity. Our approach is complementary in that we focus on correlations that can emerge in parameter dynamics during training when mini-batches are temporally dependent.

## 3. Methodology

In this section, we develop the parameter-decorrelation perspective and study how temporally dependent sampling influences training and the induced parameter distribution. Specifically, we formalize MTSF and stochastic optimiza-

tion (Section 3.1), derive a Gaussian transition model from an SDE viewpoint and introduce a scalar transition-variance signal (Section 3.2), and use a PAC-Bayes bound as a diagnostic to show how signal inflation can render complexity terms vacuous (Section 3.3). Next, we justify transition-wise KL control as a sufficient surrogate for controlling marginal KL drift along the induced parameter distribution (Section 3.4). Finally, we derive the TVA rule (Section 3.5), together with a lightweight recursion and a practical training objective (Section 3.6). Proofs of the lemmas and the theorem are provided in Appendix A.1 through Appendix A.3, and pseudo-code is provided in Appendix A.4.

### 3.1. Problem Settings

We consider MTSF, where each training example consists of a look-back window of covariates $\mathbf{x} \in \mathcal{X}$ and a look-forward horizon $\mathbf{y} \in \mathcal{Y}$ to be predicted. Let $f_\theta : \mathcal{X} \to \mathcal{Y}$ be a forecaster parameterized by $\hat{\theta} \sim \theta \in \mathbb{R}^d$, inducing a conditional model $p(\mathbf{y}|\mathbf{x}; \theta)$. We write $z = (\mathbf{x}, \mathbf{y}) \in \mathcal{Z}$ and let $S = \{z_i\}_{i=1}^n$ denote a mini-batch of size $n$. The population risk is

$$R(\theta) \triangleq \mathbb{E}_{z \sim D}\big[\ell(\theta, z)\big], \tag{1}$$

where $\ell(\theta, z) = -\log p(\mathbf{y}|\mathbf{x}; \theta)$. In practice, we minimize the empirical risk

$$L_S(\theta) \triangleq \frac{1}{n} \sum_{i=1}^n \ell(\theta, z_i) \tag{2}$$

via a stochastic optimizer. Abstractly, one update can be written as

$$\hat{\theta}_k = \hat{\theta}_{k-1} + \Delta\theta_k, \Delta\theta_k = -\eta_k \widehat{\nabla} L_{S_k}(\hat{\theta}_{k-1}), \tag{3}$$

where $S_k$ is the time-series mini-batch used at step $k$ and $\eta_k$ is the step size.

In MTSF, sliding-window sampling induces strong dependence among mini-batches, so the stochastic gradients $\widehat{\nabla} L_{S_k}$ can be temporally correlated across $k$. This temporal dependence can propagate into the parameter trajectory $\{\hat{\theta}_k\}$, reducing the effective mixing of the induced parameter distribution and increasing the risk of overfitting. Our method addresses this phenomenon by controlling parameter transitions between consecutive optimization steps.

### 3.2. Probabilistic Framework: Gaussian Transition Model and Signal for Temporal Correlation

In a small-step regime, stochastic-gradient methods admit SDE approximations that make explicit the role of gradient noise (Li et al., 2019; 2021; Malladi et al., 2022). A standard Itô form is

$$d\theta_t = -g(\theta_t)\,dt + \sqrt{\eta}\,B(\theta_t)\,dW_t, \Sigma(\theta_t) = B(\theta_t)B(\theta_t)^\top, \tag{4}$$

where $g(\theta)$ denotes a empirical gradient field, $W_t$ is a standard Brownian motion, and $\Sigma(\theta_t)$ is the local covariance of the stochastic gradient noise. Specifically, we define: $\Sigma(\theta_t) = 1/t \sum_{\tau=1}^t (g(\theta_\tau) - \bar{g}(\theta_\tau))(g(\theta_\tau) - \bar{g}(\theta_\tau))^\top$ with $\bar{g}(\theta_t) = 1/t \sum_{\tau=1}^t g(\theta_\tau)$ as the mean gradient.

The density $p(\theta, t)$ then evolves according to the Fokker–Planck–Kolmogorov equation (Särkkä & Solin, 2019)

$$\partial_t p(\theta, t) = \sum_{i=1}^d \frac{\partial}{\partial \theta^i} \big( p(\theta, t) \, [g(\theta)]_i \big) \tag{5}$$

$$+ \frac{\eta}{2} \sum_{i=1}^d \sum_{j=1}^d \frac{\partial^2}{\partial \theta^i \partial \theta^j} \big( p(\theta, t) \, [\Sigma(\theta)]_{ij} \big). \tag{6}$$

Since solving Eq. (6) is intractable for modern deep networks, we adopt a Gaussian assumed-density approximation $p(\theta, t) \approx \mathcal{N}(\theta | m_t, V_t)$ (Ansari et al., 2023). Linearizing the drift around $m_t$ gives $g(\theta) \approx g(m_t) + G_t(\theta - m_t)$ with Jacobian $G_t = \nabla_\theta g(m_t)$. The corresponding mean and covariance dynamics satisfy

$$\frac{dm_t}{dt} = -g(m_t), \frac{dV_t}{dt} = -G_t V_t - V_t G_t^\top + \eta\,\Sigma(m_t), \tag{7}$$

which is a standard consequence of Gaussian approximations for SDEs.

**Discrete-time transition kernel.** Our algorithm operates per optimizer step, so we adopt a discrete-time transition model. Decompose the stochastic gradient into its conditional mean plus noise:

$$\widehat{\nabla} L_{S_k}(\theta) = g_k(\theta) + \xi_k, \mathbb{E}[\xi_k|\theta] = 0, \mathrm{Cov}(\xi_k|\theta) = \Sigma_k(\theta), \tag{8}$$

where $g_k(\theta)$ is the conditional mean gradient under the mini-batch sampling mechanism at step $k$, and $\Sigma_k(\theta)$ is the conditional gradient-noise covariance. Substituting Eq. (8) into the update rule Eq. (3) yields the Gaussian transition approximation, conditional on $\hat{\theta}_{k-1}$,

$$\hat{\theta}_k | \hat{\theta}_{k-1} \approx \mathcal{N}\Big(\hat{\theta}_{k-1} - \eta_k g_k(\hat{\theta}_{k-1}),\ \eta_k^2 \Sigma_k(\hat{\theta}_{k-1})\Big). \tag{9}$$

Eq. (9) makes explicit that the effective step size $\eta_k$ controls the transition covariance quadratically, which provides a direct lever for regulating the uncertainty of parameter moves across steps.

**Transition-variance signal.** Estimating the full matrix $\Sigma_k(\hat{\theta}_{k-1})$ is typically infeasible, so we track its mean-field scale $\sigma_k^2 \triangleq \frac{1}{d}\mathrm{tr}\big(\Sigma_k(\hat{\theta}_{k-1})\big)$, and approximate $\Sigma_k \approx \sigma_k^2 I$. With this approximation, Eq. (9) simplifies to

$$p(\theta_k | \theta_{k-1}) = \mathcal{N}\Big(\theta_k | \theta_{k-1} - \eta_k g_k(\theta_{k-1}),\ \sigma_k^2 \eta_k^2 I\Big). \tag{10}$$

where the scalar $\sigma_k^2 \eta_k^2$ serves as our transition-variance signal. This quantity measures the scale of step-to-step update

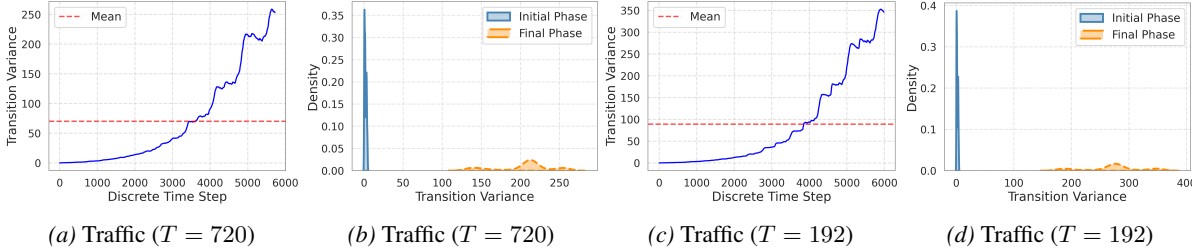

*(a)* Traffic ($T = 720$)     *(b)* Traffic ($T = 720$)     *(c)* Traffic ($T = 192$)     *(d)* Traffic ($T = 192$)

*Figure 1.* Transition-variance dynamics and their distribution in the Traffic dataset. Subfigures (a) and (c) illustrate the variance dynamics over discrete time steps, while (b) and (d) present kernel density estimation plots of the transition-variance computed from the first 20% (*Initial Phase*) and last 20% (*Final Phase*) segments. The look-back window $L$ is fixed to 96, with prediction lengths $T$ of 720 for (a, b) and 192 for (c, d). In both long- and short-horizon forecasting settings, the transition-variance increases progressively, indicating non-stationary behavior. Additional results on various datasets are presented in Appendix C.1.

uncertainty; in MTSF, temporally persistent gradient-noise effects induced by overlapping windows can cause this signal to vary non-trivially over training, thereby providing a practical handle on optimization-induced dependence in the parameter trajectory.

### 3.3. Diverging Transition-Variance and a PAC-Bayes Diagnostic

Empirically (see Fig. 1), the transition-variance signal increase over training time step in both long-horizon and shorter-horizon TSF tasks. Persistent inflation of this quantity suggests that the induced parameter distribution spreads over time and drifts away from any fixed reference distribution. As a consequence, PAC-Bayes generalization certificates—whose complexity term depends on a KL divergence between a data-dependent posterior and a fixed prior—can become vacuous.

We adopt the PAC-Bayes inequality stated in a convenient $\lambda$-form for bounded losses (Seeger, 2002; Alquier, 2021; Rothfuss et al., 2023). Let $\rho$ be a data-dependent posterior distribution over parameters and $\pi$ a fixed prior independent of the sample $S$. Assume $\ell(\theta, z) \in [0, C]$ for all $(\theta, z)$.

**Lemma 3.1** (PAC-Bayes certificate worsens under diverging posterior variance). *Fix $\delta \in (0, 1)$ and any $\lambda > 0$ chosen independently of $S$. With probability at least $1 - \delta$ over $S \sim D^n$, simultaneously for all distributions $\rho$ on $\mathbb{R}^d$,*

$$\mathbb{E}_{\theta \sim \rho}\big[R(\theta)\big] \leq \mathbb{E}_{\theta \sim \rho}\big[L_S(\theta)\big] \tag{11}$$

$$+ \frac{\lambda C^2}{8n} + \frac{\mathrm{KL}(\rho \| \pi) + \log(1/\delta)}{\lambda}. \tag{12}$$

*Assume $\pi = \mathcal{N}(0, \sigma^2 I)$ and $\rho_k = \mathcal{N}(m_k, s_k^2 I)$, where $(s_k^2)_{k \geq 1}$ is nondecreasing and diverges ($s_k^2 \to \infty$) while $(\|m_k\|_2)_{k \geq 1}$ is bounded. Then, for any fixed $\lambda > 0$, the PAC-Bayes complexity term*

$$\mathcal{C}_k(\lambda) \triangleq \frac{\mathrm{KL}(\rho_k \| \pi) + \log(1/\delta)}{\lambda}$$

*is eventually nondecreasing and satisfies $\mathcal{C}_k(\lambda) \to \infty$ as $k \to \infty$. Consequently, the bound (12) worsens for all*

*sufficiently large $k$ and becomes vacuous along any such variance-diverging trajectory.*

Lemma 3.1 motivates stabilizing the variance of the induced parameter distribution. In our framework, this is achieved by regulating the transition-variance appearing in Eq. (10).

### 3.4. Parameter Decorrelation via Transition-KL Control

We focus on the PAC-Bayes complexity term, which depends on a marginal KL divergence (e.g., $\mathrm{KL}(\rho_k \| \pi)$). Our key move is to control an upper bound on a marginal KL by a sum of transition KL divergences between Markov kernels. This motivates a local control strategy in which keeping each step-to-step transition close to a reference transition prevents the induced marginal distribution from drifting too far from the corresponding reference marginal.

Let $(P_k)_{k \geq 1}$ and $(Q_k)_{k \geq 1}$ be Markov kernels on $\mathbb{R}^d$. Starting from initial distributions $p_0$ and $q_0$, define the induced marginals recursively by

$$p_k(\cdot) = \int P_k(\cdot | \theta) \, p_{k-1}(d\theta), \quad q_k(\cdot) = \int Q_k(\cdot | \theta) \, q_{k-1}(d\theta).$$

**Lemma 3.2** (Transition KL control implies marginal KL control). *For all $k \geq 1$,*

$$\mathrm{KL}(p_k \| q_k) \leq \mathrm{KL}(p_0 \| q_0)$$

$$+ \sum_{i=0}^{k-1} \mathbb{E}_{\theta_i \sim p_i} \left[\mathrm{KL}\left(P_{i+1}(\cdot | \theta_i) \| Q_{i+1}(\cdot | \theta_i)\right)\right]. \tag{13}$$

*In particular, if $p_0 = q_0$, then reducing the expected transition KL terms tightens an upper bound on $\mathrm{KL}(p_k \| q_k)$.*

Lemma 3.2 justifies a local control strategy by minimizing the expected transition KL between a data-driven kernel $P_k$ and a reference kernel $Q_k$; we control an upper bound on the marginal KL drift along the induced parameter distribution.

## 3.5. Transition-Variance Alignment (TVA)

Motivated by the isotropic transition kernel in Eq. (10), we define a data-driven transition kernel

$$P_k(\cdot|\theta_{k-1}) = \mathcal{N}\Big( \cdot \ \big|\ \theta_{k-1} - \eta_k g_k(\theta_{k-1}),\ \sigma_k^2\eta_k^2 I \Big), \quad (14)$$

and a reference kernel with a target transition-variance $\sigma_\lambda^2\eta^2$:

$$Q_k(\cdot|\theta_{k-1}) = \mathcal{N}\Big( \cdot \ \big|\ \theta_{k-1} - \eta_k g_k(\theta_{k-1}),\ \sigma_\lambda^2\eta^2 I \Big). \quad (15)$$

We match the drift so that the transition KL depends only on covariance mismatch, yielding a stable control signal. For Gaussians with matched means and isotropic covariances, letting $s_k^2 = \sigma_k^2\eta_k^2$ and $u^2 = \sigma_\lambda^2\eta^2$, the transition KL admits the closed form

$$\mathrm{KL}(P_k\|Q_k) = \frac{d}{2}\left( \frac{s_k^2}{u^2} - 1 - \log\frac{s_k^2}{u^2} \right). \quad (16)$$

**Controlling the transition-variance via a step-size gate.** We parameterize the step size as

$$\eta_k = \alpha_k\,\eta, \quad (17)$$

where $\alpha_k \in (0,1]$ and $\eta$ is a baseline step size. The data-driven transition-variance becomes $s_k^2 = \sigma_k^2\alpha_k^2\eta^2$. To prevent variance blow-up while keeping the update smooth, we use the symmetric log-ratio discrepancy rule

$$\alpha_k \triangleq \exp\Big( - \big| \log(\sigma_k + \varepsilon) - \log\sigma_\lambda \big| \Big)$$
$$= \min\left\{ \frac{\sigma_k + \varepsilon}{\sigma_\lambda},\ \frac{\sigma_\lambda}{\sigma_k + \varepsilon} \right\}, \quad (18)$$

with a small $\varepsilon > 0$ for numerical stability. With Eq. (17)–(18), the controlled transition becomes

$$p(\theta_k|\theta_{k-1}) = \mathcal{N}\Big( \theta_k \ \big|\ \theta_{k-1} - \alpha_k\eta\, g_k(\theta_{k-1}),\ \sigma_k^2(\alpha_k\eta)^2 I \Big). \quad (19)$$

This gate has three useful properties. First, $\alpha_k \leq 1$ always and $\alpha_k \approx 1$ when $\sigma_k \approx \sigma_\lambda$. Second, in the early phase where $\sigma_k$ is small relative to $\sigma_\lambda$, the rule behaves as $\alpha_k \approx (\sigma_k + \varepsilon)/\sigma_\lambda$, which increases the effective transition-variance smoothly as training proceeds. Third, as the mismatch grows on the log scale, $\alpha_k$ decreases smoothly; since $\alpha_k = \exp(-|\Delta_k|)$ with $\Delta_k = \log(\sigma_k + \varepsilon) - \log\sigma_\lambda$, the mapping $\Delta \mapsto \exp(-|\Delta|)$ is globally Lipschitz, which helps avoid abrupt changes in the controlled transition-variance and yields stable training dynamics.

The following theorem formalizes that the gate in Eq. (18) prevents variance-driven blow-up by uniformly bounding the per-step transition KL when $\sigma_k$ becomes large.

**Theorem 3.3** (Complexity control via adaptive transition-variance alignment)**.** *Fix $\delta \in (0,1)$ and assume the bounded-loss setting of Lemma 3.1. Let $\{\theta_k\}_{k\geq 0}$ be the Markov chain induced by $P_k(\cdot|\theta_{k-1})$ in Eq. (14) with gated step size $\eta_k = \alpha_k\eta$ and $\alpha_k$ given by Eq. (18). Let $\{q_k\}_{k\geq 0}$ be a reference chain induced by $Q_k(\cdot|\theta_{k-1})$ in Eq. (15), and denote their marginals by $p_k$ and $q_k$. Assume $p_0 = q_0$. Then for every $k \geq 1$,*

$$\mathrm{KL}(p_k\|q_k) \leq \sum_{i=0}^{k-1} \mathbb{E}_{\theta_i \sim p_i}\left[ \mathrm{KL}\left( P_{i+1}(\cdot|\theta_i)\|Q_{i+1}(\cdot|\theta_i) \right) \right]. \quad (20)$$

*Moreover, since $P_i$ and $Q_i$ have matched means and isotropic covariances,*

$$\mathrm{KL}(P_i\|Q_i) = \frac{d}{2}\left( r_i - 1 - \log r_i \right), \quad (21)$$

*where $r_i \triangleq \frac{\sigma_i^2\alpha_i^2}{\sigma_\lambda^2}$. Under the gate Eq. (18), we always have $\sigma_i\alpha_i \leq \sigma_\lambda$ and hence $r_i \leq 1$. In particular, in the variance-inflation regime $\sigma_i \geq \sigma_\lambda$, the gate yields $\alpha_i = \sigma_\lambda/(\sigma_i + \varepsilon)$ and therefore*

$$r_i = \left( \frac{\sigma_i}{\sigma_i + \varepsilon} \right)^2 \in \left[ (\tfrac{\sigma_\lambda}{\sigma_\lambda + \varepsilon})^2, 1 \right), \quad (22)$$

*so $\mathrm{KL}(P_i\|Q_i)$ is uniformly bounded over this regime. Consequently, TVA prevents variance-driven explosion of the transition-KL contributions when $\sigma_i$ becomes large, and thus controls an upper bound on marginal KL drift via Eq. (20).*

## 3.6. Recursion and Training Objective

The controlled transition Eq. (19) enables a recursive approximation to the marginal parameter distribution. Let $p(\theta_k) = \mathcal{N}(\theta_k|m_k, V_k)$ denote the marginal at step $k$. By the Chapman–Kolmogorov equation (Särkkä & Svensson, 2023),

$$p(\theta_k) = \int p(\theta_k|\theta_{k-1})\, p(\theta_{k-1})\, d\theta_{k-1}, \quad (23)$$

and noting that both the marginal and transition are Gaussian under our approximation, we obtain the recursion

$$m_k = m_{k-1} - \alpha_k\eta\, g_k(m_{k-1}), \quad V_k = V_{k-1} + \sigma_k^2(\alpha_k\eta)^2 I, \quad (24)$$

with $m_0 = \hat{\theta}_0$ and $V_0 = 0$. This representation separates mean dynamics from covariance dynamics, and makes explicit how the TVA gate $\alpha_k$ regulates the per-step increase in marginal variance.

**Prediction and loss.** Exact Bayesian prediction would average the predictive distribution over $p(\theta_k)$:

$$p(\mathbf{y}|\mathbf{x}; m_k) \approx \int p(\mathbf{y}|\mathbf{x}, \theta)\, p(\theta_k)(d\theta). \quad (25)$$

*Table 1.* Multivariate time-series forecasting results with prediction length $T \in \{96, 192, 336, 720\}$. Results are averaged over all prediction lengths. ECL and WTH denote the Electricity and Weather datasets, respectively. The best and second-best results are highlighted in **bold** and underlined, respectively.

| Model | MLP | | MLP+TVA | | iTransformer | | PatchTST | | MSGNet | | Crossformer | | TimesNet | |
|---|---|---|---|---|---|---|---|---|---|---|---|---|---|---|
| Metric | MSE | MAE | MSE | MAE | MSE | MAE | MSE | MAE | MSE | MAE | MSE | MAE | MSE | MAE |
| ETTh1 | 0.448 | 0.435 | **0.438** | **0.432** | 0.449 | 0.442 | 0.451 | 0.439 | 0.469 | 0.465 | 0.580 | 0.554 | 0.468 | 0.458 |
| ETTh2 | 0.374 | 0.400 | **0.371** | **0.397** | 0.379 | 0.406 | 0.395 | 0.412 | 0.393 | 0.418 | 0.805 | 0.642 | 0.406 | 0.423 |
| ETTm1 | 0.393 | 0.399 | **0.389** | 0.398 | 0.413 | 0.413 | 0.390 | **0.397** | 0.425 | 0.422 | 0.502 | 0.511 | 0.409 | 0.416 |
| ETTm2 | 0.290 | 0.335 | **0.278** | **0.324** | 0.293 | 0.337 | 0.288 | 0.334 | 0.291 | 0.331 | 0.482 | 0.487 | 0.296 | 0.333 |
| ECL | 0.218 | 0.299 | 0.195 | 0.285 | **0.190** | **0.280** | 0.259 | 0.346 | 0.337 | 0.410 | 0.327 | 0.398 | 0.238 | 0.331 |
| Solar | 0.277 | 0.352 | **0.234** | **0.310** | 0.298 | 0.315 | 0.349 | 0.369 | 0.625 | 0.521 | 0.272 | 0.311 | 0.316 | 0.316 |
| Traffic | 0.508 | 0.327 | 0.452 | 0.300 | **0.435** | **0.289** | 0.554 | 0.351 | 0.656 | 0.355 | 0.598 | 0.329 | 0.645 | 0.344 |
| WTH | 0.273 | 0.291 | **0.260** | **0.280** | 0.275 | 0.293 | 0.280 | 0.297 | 0.265 | 0.290 | 0.384 | 0.431 | 0.267 | 0.290 |

which is generally intractable for modern forecasters. For efficiency, we adopt a standard plug-in approximation,

$$\ell\big(m_k, (\mathbf{x}, \mathbf{y})\big) = -\log p(\mathbf{y}|\mathbf{x}; m_k). \qquad (26)$$

**Training objective with TVA.** At iteration $k$, TVA computes the gate $\alpha_k$ via Eq. (18) and performs the gated update Eq. (3) with $\eta_k = \alpha_k \eta$. The method therefore modifies the optimization trajectory through $\alpha_k$ while directly regulating the transition-variance $\sigma_k^2(\alpha_k \eta)^2$ in Eq. (19). Following Theorem 3.3, this yields a principled mechanism to prevent variance-driven blow-up of the transition-KL surrogate and to stabilize the induced parameter distribution over training.

## 4. Experiments

In this section, we investigated the effectiveness and efficiency of TVA on standard MTSF benchmarks, comparing against widely used architectures and existing techniques. Additional analyses—including the evolution of the transition-variance under TVA, visualizations of forecasts, robustness and reproducibility, and full results for all tables—are provided in Appendix C.

### 4.1. Settings

**Dataset and Metrics.** We used eight widely adopted time-series forecasting datasets—the ETT series (ETTh1, ETTh2, ETTm1, ETTm2) (Zhou et al., 2021), Weather, Traffic, Electricity, and Solar (Wu et al., 2021)—to evaluate the proposed method. A detailed description of each dataset is provided in Appendix B.1. Following prior studies (Zhou et al., 2021; Wu et al., 2021; Zhou et al., 2022), we reported mean squared error (MSE) and mean absolute error (MAE). MSE penalizes larger deviations more heavily, while MAE measures average absolute prediction error; both metrics are standard in TSF benchmarks for their interpretability and widespread use.

**Baselines.** To assess forecasting accuracy relative to established methods, we compared against representative

Transformer-based forecasters, including iTransformer (Liu et al., 2024), PatchTST (Nie et al., 2023), MSGNet (Cai et al., 2024), and Crossformer (Zhang & Yan, 2023), as well as the TCN-based TimesNet (Wu et al., 2023). We additionally included DLinear (Zeng et al., 2023), CLinear (Lin et al., 2024a), LDLinear (Yu et al., 2024) as a strong and widely used linear baseline.

**Implementation Details.** The proposed TVA method is architecture-agnostic and can be integrated into diverse forecasting backbones by modifying only the optimizer step size according to the gate. Unless otherwise stated, we used a lightweight MLP as the primary backbone for controlled evaluations. The architecture follows the MLP design in (Lin et al., 2024a): a channel-wise input projection layer, two feedforward layers with Gaussian error linear unit (GELU) activations (Hendrycks & Gimpel, 2016), and an output layer mapping hidden representations to the prediction horizon. Additional architectural details are provided in Appendix B.2. All models were evaluated under the same experimental protocol across datasets, with prediction lengths $T \in \{96, 192, 336, 720\}$ and a fixed look-back window $L = 96$. We followed common preprocessing procedures used in prior works (e.g., iTransformer (Liu et al., 2024), PatchTST (Nie et al., 2023)), including dataset splitting and normalization. Models were trained using Adam (Kingma & Ba, 2015), and the random seed was fixed to 2026 to ensure reproducibility. The TVA target scale was selected by a greedy search over $\sigma_\lambda \in \{1.0, 1.5, \ldots, 10.0\}$ and $\epsilon$ to $1 \times 10^{-8}$. All experiments were conducted on a single NVIDIA GeForce RTX 4090 GPU.

### 4.2. Main Results

Table 1 summarizes multivariate forecasting performance, comparing the TVA-enhanced MLP with state-of-the-art Transformer- and TCN-based baselines. Incorporating TVA into the MLP backbone consistently improved forecasting accuracy across datasets and prediction horizons, reducing the average MSE by approximately 1% to over 15%. On the Solar dataset, the MLP with TVA achieved an MSE of

*Table 2.* Comparison of different normalization techniques for diverse models, with and without the TVA method. Results are averaged over all prediction lengths. +TVA indicates whether TVA is applied. The x-mark denotes the normalization method alone, while the check-mark indicates its combination with TVA. The lower MSE values are highlighted in **bold**.

| Models | iTransformer | | | | | | DLinear | | | | | | MLP | | | | | |
|---|---|---|---|---|---|---|---|---|---|---|---|---|---|---|---|---|---|---|
| Norm. | RevIN | | DishTS | | SAN | | RevIN | | DishTS | | SAN | | RevIN | | DishTS | | SAN | |
| +TVA | ✗ | ✓ | ✗ | ✓ | ✗ | ✓ | ✗ | ✓ | ✗ | ✓ | ✗ | ✓ | ✗ | ✓ | ✗ | ✓ | ✗ | ✓ |
| ETTh1 | 0.449 | **0.443** | 0.484 | **0.470** | 0.657 | **0.529** | **0.446** | **0.446** | 0.477 | **0.468** | 0.651 | **0.511** | 0.448 | **0.438** | **0.484** | 0.487 | 0.544 | **0.469** |
| ETTh2 | 0.379 | **0.378** | 0.816 | **0.812** | 0.406 | **0.404** | **0.374** | 0.377 | 0.636 | **0.596** | **0.403** | **0.403** | 0.374 | **0.371** | **0.548** | **0.548** | 0.400 | **0.398** |
| ETTm1 | 0.413 | **0.410** | 0.421 | **0.415** | 0.493 | **0.463** | 0.407 | **0.399** | **0.411** | 0.428 | 0.466 | **0.415** | 0.393 | **0.389** | 0.420 | **0.416** | 0.437 | **0.405** |
| ETTm2 | 0.293 | **0.289** | 0.409 | **0.390** | **0.291** | **0.291** | 0.289 | **0.283** | 0.357 | **0.336** | **0.291** | **0.291** | 0.290 | **0.278** | **0.310** | 0.312 | 0.303 | **0.291** |
| Electricity | 0.190 | **0.187** | **0.214** | 0.247 | **0.185** | 0.196 | 0.221 | **0.207** | **0.276** | **0.276** | 0.223 | **0.217** | 0.218 | **0.195** | **0.238** | 0.270 | 0.202 | **0.189** |
| Solar | 0.298 | **0.276** | 0.275 | **0.256** | 0.266 | **0.256** | 0.352 | **0.269** | 0.330 | **0.281** | 0.303 | **0.251** | 0.277 | **0.234** | 0.279 | **0.264** | 0.256 | **0.250** |
| Traffic | **0.435** | 0.445 | **0.448** | 0.474 | 0.489 | **0.480** | 0.630 | **0.503** | 0.671 | **0.585** | 0.614 | **0.550** | 0.508 | **0.452** | 0.551 | **0.535** | 0.508 | **0.493** |
| Weather | 0.275 | **0.270** | 0.365 | **0.363** | 0.275 | **0.270** | 0.273 | **0.270** | **0.299** | 0.335 | **0.265** | 0.269 | 0.273 | **0.260** | **0.313** | 0.334 | 0.267 | **0.265** |

*Table 3.* Comparison of different STD methods with and without TVA. Results are averaged over all prediction lengths. The x-mark denotes the STD method alone, while the check-mark indicates its combination with TVA. The lower MSE values are highlighted in **bold**.

| Models | CLinear | | LDLinear | | DLinear | | Linear | |
|---|---|---|---|---|---|---|---|---|
| +TVA | ✗ | ✓ | ✗ | ✓ | ✗ | ✓ | ✗ | ✓ |
| ETTh1 | 0.444 | **0.442** | 0.435 | **0.433** | **0.446** | **0.446** | **0.447** | **0.447** |
| ETTh2 | 0.392 | **0.384** | 0.375 | **0.369** | **0.374** | 0.377 | **0.375** | 0.382 |
| ETTm1 | 0.394 | **0.389** | 0.392 | **0.388** | 0.407 | **0.399** | 0.407 | **0.400** |
| ETTm2 | 0.295 | **0.272** | 0.287 | **0.280** | 0.289 | **0.283** | 0.298 | **0.283** |
| ECL | 0.185 | **0.177** | 0.191 | **0.189** | 0.221 | **0.207** | 0.219 | **0.206** |
| Solar | 0.312 | **0.249** | 0.288 | **0.263** | 0.352 | **0.269** | 0.343 | **0.263** |
| Traffic | 0.483 | **0.477** | **0.447** | 0.449 | 0.630 | **0.503** | 0.629 | **0.503** |
| WTH | 0.258 | **0.252** | 0.262 | **0.261** | 0.273 | **0.270** | 0.272 | **0.270** |

*Table 4.* Efficiency comparison on the Electricity dataset with look-back window $L = 96$ and prediction length $T = 720$.

| Model | Parameters | Time (ms) | MSE↓ |
|---|---|---|---|
| Linear | 69.8K | 8.0 | 0.266 |
| Linear + TVA | 69.8K | 9.0 | **0.249** |
| DLinear | 139.7K | 8.2 | 0.270 |
| DLinear + TVA | 139.7K | 9.3 | **0.251** |
| MLP | 0.94M | 7.8 | 0.274 |
| MLP + TVA | 0.94M | 9.2 | **0.253** |
| iTransformer | 5.15M | 43.4 | 0.275 |
| iTransformer + TVA | 5.15M | 51.9 | **0.244** |

0.234, corresponding to an improvement of approximately 16% over the vanilla MLP. These results indicate that TVA functions as an effective parameter-level regularizer, mitigating overfitting through parameter dynamics control without architectural modifications or additional trainable parameters. Despite its structural simplicity, the TVA-enhanced MLP is frequently competitive with, and sometimes superior to, higher-capacity models. It achieved the lowest MSE on the ETT series, Solar, and Weather benchmarks among the considered baselines. On the remaining datasets, the method consistently ranked first or second in terms of both MSE and MAE, demonstrating stable forecasting behavior under varying sampling frequencies. Overall, these findings highlight that combining a minimal backbone with TVA yields strong predictive accuracy and robustness while maintaining substantially lower computational complexity than modern large-scale forecasting architectures.

### 4.3. Analysis

**Combining TVA with instance normalization methods for diverse models.** As shown in Table 2, TVA consistently improved forecasting accuracy across diverse backbones, including iTransformer, DLinear, and MLP, under multiple normalization schemes. For example, on the Solar dataset, adding TVA yielded an average 10.8% reduction in MSE across the tested configurations compared to using the corresponding normalization method alone. These results support two conclusions. First, TVA is architecture-agnostic. The observed gains are not tied to a specific architecture or normalization pipeline. Second, TVA complements data-level normalization. Whereas instance normalization primarily reduces nuisance variation and correlation in the input representation, TVA acts at the parameter level by regulating the transition-variance from optimization.

**Compatibility of TVA with STD techniques.** Table 3 compares several STD techniques with and without TVA, and also includes a vanilla Linear model as a reference baseline. As shown in Table 3, adding TVA to the Linear model often matched or even outperformed more complex STD-based architectures that do not incorporate parameter dynamics control. For instance, on the Solar dataset, Linear+TVA attained an average MSE of 0.263, improving upon the standard versions of CLinear (0.312), LDLinear (0.288), and DLinear (0.352). TVA also remained compatible with STD-based pipelines, since integrating TVA into CLinear, LDLinear, or DLinear typically yields additional gains relative to the corresponding STD method alone. Overall, STD and TVA address complementary failure modes.

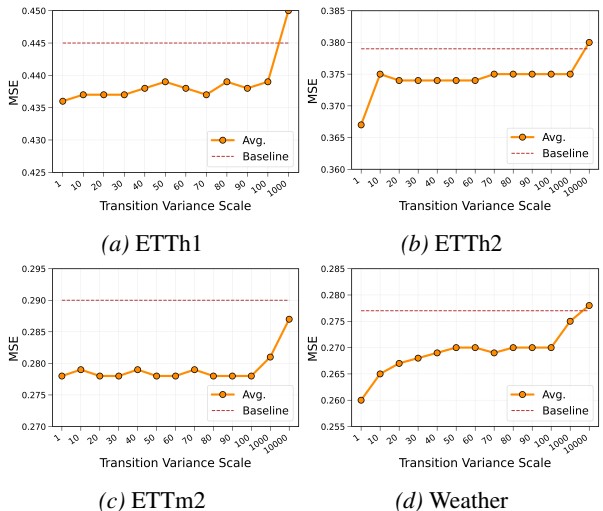

*Figure 2.* Performance of the MLP model with different transition-variance scales. Results are averaged over all prediction lengths. Dotted lines represent the baseline performance without TVA.

STD stabilizes the input by separating predictable components, while TVA mitigates optimization-induced overfitting by regulating the transition-variance during training. Combining stabilized representations with controlled parameter dynamics, therefore, improves generalization. This synergy is reflected in the generally improved performance observed when TVA is enabled (check-marks) compared to the corresponding STD-only configurations (x-marks).

**Efficiency.** TVA is computationally lightweight, as it introduces only a scalar gating operation that modulates the optimizer step size, without requiring any architectural modifications or additional trainable parameters. As reported in Table 4, TVA incurs only a modest increase in per-batch runtime, ranging from 1.0 ms to 1.4 ms for lightweight models and 8.5 ms for iTransformer. Despite this minimal overhead, TVA reduced MSE across all backbones. For example, Linear and DLinear models achieved noticeable error reductions, while the TVA-enhanced MLP improved MSE from 0.274 to 0.253. In addition, the iTransformer augmented with TVA reduced MSE from 0.275 to 0.244, demonstrating that TVA remains effective even when applied to high-capacity models. Overall, these results indicate a favorable trade-off between accuracy and efficiency. TVA operates at the optimization level without introducing architectural modifications or additional trainable parameters, making it a practical and scalable regularization strategy for MTSF.

**Sensitivity Analysis.** The hyperparameter $\sigma_\lambda$ sets the target scale of the transition-variance and therefore controls the regularization strength in TVA. To assess sensitivity, we varied $\sigma_\lambda$ over a broad range. Figure 2 reports MSE on ETTh1, ETTh2, ETTm2, and Weather, averaged over all prediction lengths $T \in \{96, 192, 336, 720\}$ with a fixed look-back win-

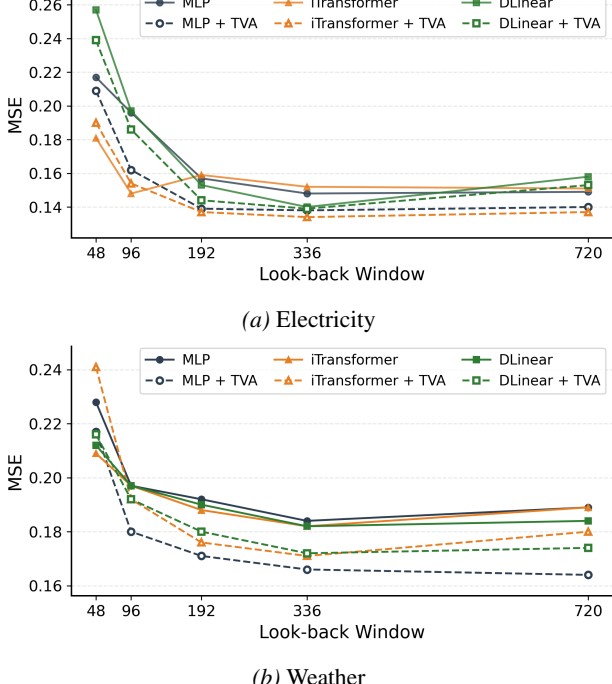

*(a)* Electricity

*(b)* Weather

*Figure 3.* Performance of the MLP model under different look-back windows $L \in \{48, 96, 192, 336, 720\}$ on Electricity and Weather datasets. The prediction length $T$ is fixed to 96.

dow $L = 96$. TVA exhibited stable performance across a wide spectrum of target scales. On ETTh1, MSE remained consistently below the baseline for $\sigma_\lambda$ values up to 100, with noticeable degradation only at an extreme value of 1000. On ETTm2, performance remained stable up to 1000 and continues to outperform the baseline even at 10000. Although ETTh2 and Weather showed comparatively higher sensitivity, TVA remained robust over a practically relevant range of $\sigma_\lambda$. These findings suggest that TVA can stabilize parameter dynamics without requiring exhaustive hyperparameter tuning. In practice, the method is robust to the choice of $\sigma_\lambda$, substantially reducing the burden of hyperparameter search and improving applicability across diverse scenarios.

**Performance on Various Look-back Window.** Longer look-back windows can introduce redundant history, amplify noise, and increase exposure to non-stationary shifts, which often makes optimization harder and degrades generalization (Zeng et al., 2023; Liu et al., 2023). We therefore evaluated TVA under varying look-back windows $L \in \{48, 96, 192, 336, 720\}$ while fixing the prediction length to $T = 96$. As shown in Figure 3, the baseline MLP exhibited the typical trend that performance deteriorated as $L$ became large, whereas the TVA-enhanced MLP remained noticeably more stable. TVA generally improved accuracy across the tested $L$, with the gains becoming more apparent in the long-context regime, where redundant or weakly informative history most strongly harms the baseline. This

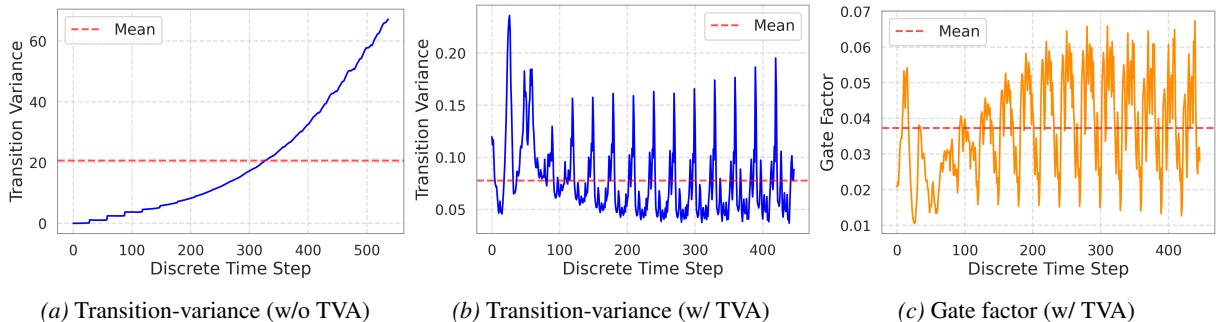

*(a)* Transition-variance (w/o TVA)    *(b)* Transition-variance (w/ TVA)    *(c)* Gate factor (w/ TVA)

*Figure 4.* Transition-variance dynamics with and without TVA, together with the corresponding adaptive gate behavior on the ETTh1 dataset. The look-back window $L$ is fixed to 96, and the prediction length $T$ is fixed to 720. Additional results on various datasets are presented in Appendix C.1.

pattern is consistent with TVA's role as a parameter-level regularizer. Longer look-back windows induce stronger dependence among sliding-window mini-batches and more persistent gradient noise, which can inflate the transition-variance and yield overly correlated parameter updates. By damping such unstable updates, it mitigates the performance degradation that typically arises when overly long context is used.

**Transition-Variance Dynamics.** To examine whether TVA actively regulates transition-variance dynamics during training, we visualized the transition-variance trajectories together with the adaptive gate behavior. As shown in Figure 4(a), without TVA, the transition-variance progressively increased throughout training, indicating unstable parameter transitions and amplified update dynamics. In contrast, Figure 4(b) demonstrates that TVA kept the transition-variance bounded, with fluctuations concentrated around a stable regime. This bounded behavior is consistent with the theoretical motivation in Section 3, where controlling transition-variance mitigates optimization-induced dependence and prevents the accumulation of excessive temporal dependencies in parameter dynamics. Figure 4(c) further illustrates that the adaptive gate factor $\alpha_k$ varied dynamically rather than remaining fixed throughout training. Several pronounced decreases in $\alpha_k$ were observed, which were consistent with periods where temporary transition-variance increases appeared in Figure 4(b). These observations suggest that TVA constrains transition-variance growth through adaptive step-size modulation, rather than by uniformly reducing update magnitudes. Additional results on various datasets are provided in Appendix C.1.

## 5. Conclusion

We studied multivariate time-series forecasting from the parameter-decorrelation perspective, complementing prior work that primarily focuses on data decorrelation. By modeling stochastic optimization as a parameter-space Markov process, we derived the transition-variance signal that cap-

tures optimization-induced dependence in the parameter dynamics, and showed that inflation of this signal can degrade generalization. To mitigate this effect, we proposed TVA, a architecture-agnostic method that controls the transition-variance during training via a smooth step-size gate. Extensive experiments on real-world benchmarks demonstrate that TVA consistently improves forecasting accuracy across diverse architectures with negligible computational overhead, while integrating seamlessly with existing approaches. The consistent gains observed when TVA is combined with the data decorrelation techniques further highlight the complementary value of regulating optimization-induced dependence. Overall, our findings emphasize the importance of controlling parameter dynamics for improved generalization in multivariate time-series forecasting.

## Acknowledgements

This work was partly supported by Institute of Information & communications Technology Planning & Evaluation (IITP) grant funded by the Korea government(MSIT) (No.RS-2020-II201373, Artificial Intelligence Graduate School Program(Hanyang University)) and the National Research Foundation of Korea(NRF) grant funded by the Korea government(MSIT) (RS-2025-00557944).

## Impact Statement

This paper presents work whose goal is to advance the field of Machine Learning. There are many potential societal consequences of our work, none which we feel must be specifically highlighted here.

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

# A. More Details of TVA

For completeness, we provide proof and additional justification for the proposed method.

## A.1. Proof of Lemma 3.1

*Proof of Lemma 3.1.* We prove the PAC-Bayes inequality and then show that a variance-diverging Gaussian posterior makes the resulting complexity term diverge.

**A $\lambda$-form PAC-Bayes inequality (Alquier, 2021).** Fix any $\lambda > 0$. For a fixed $\theta \in \mathbb{R}^d$, define $\ell_i(\theta) \triangleq \ell(\theta, z_i) \in [0, C]$, so that $L_S(\theta) = \frac{1}{n} \sum_{i=1}^n \ell_i(\theta)$ and $R(\theta) = \mathbb{E}_{z \sim D}[\ell(\theta, z)]$. By Hoeffding's lemma applied to the bounded random variable $\ell(\theta, z) - R(\theta) \in [-R(\theta), C - R(\theta)]$ (whose range length is at most $C$), and using independence across $z_1, \ldots, z_n$, we obtain

$$\mathbb{E}_{S \sim D^n}\Big[\exp\Big(\lambda\big(R(\theta) - L_S(\theta)\big)\Big)\Big] \leq \exp\Big(\frac{\lambda^2 C^2}{8n}\Big). \tag{27}$$

Integrating Eq. (27) over a prior $\pi$ and applying Fubini's theorem yields

$$\mathbb{E}_{S \sim D^n}\Big[\mathbb{E}_{\theta \sim \pi} \exp\Big(\lambda(R(\theta) - L_S(\theta))\Big)\Big] \leq \exp\Big(\frac{\lambda^2 C^2}{8n}\Big). \tag{28}$$

By Markov's inequality, with probability at least $1 - \delta$ over $S \sim D^n$,

$$\mathbb{E}_{\theta \sim \pi} \exp\Big(\lambda(R(\theta) - L_S(\theta))\Big) \leq \exp\Big(\frac{\lambda^2 C^2}{8n}\Big) \frac{1}{\delta}. \tag{29}$$

On the event Eq. (29), apply the Donsker–Varadhan change-of-measure inequality: for any posterior $\rho$ and any measurable $f$,

$$\mathbb{E}_{\theta \sim \rho}[f(\theta)] \leq \mathrm{KL}(\rho \| \pi) + \log \mathbb{E}_{\theta \sim \pi}\Big[e^{f(\theta)}\Big].$$

Choosing $f(\theta) = \lambda(R(\theta) - L_S(\theta))$ and using Eq. (29) gives

$$\lambda\Big(\mathbb{E}_{\theta \sim \rho}[R(\theta)] - \mathbb{E}_{\theta \sim \rho}[L_S(\theta)]\Big) \leq \mathrm{KL}(\rho \| \pi) + \frac{\lambda^2 C^2}{8n} + \log\Big(\frac{1}{\delta}\Big).$$

Rearranging yields Eq. (12). The bound holds for any fixed $\lambda > 0$; if one wishes to choose $\lambda$ data-dependently, use a discrete grid plus union bound.

**Diverging posterior variance makes the complexity term diverge.** Let $\pi = \mathcal{N}(0, \sigma^2 I)$ and $\rho_k = \mathcal{N}(m_k, s_k^2 I)$. The KL divergence between isotropic Gaussians admits the closed form

$$\mathrm{KL}(\rho_k \| \pi) = \frac{1}{2}\left(\frac{\|m_k\|_2^2}{\sigma^2} + d\left(\frac{s_k^2}{\sigma^2} - 1 - \log \frac{s_k^2}{\sigma^2}\right)\right). \tag{30}$$

By assumption, $\|m_k\|_2$ is bounded, so $\|m_k\|_2^2/\sigma^2 = O(1)$. Let $t_k \triangleq s_k^2/\sigma^2$. Since $(s_k^2)_{k \geq 1}$ is nondecreasing and $s_k^2 \to \infty$, we have $t_k \to \infty$ and eventually $t_k \geq 1$. Define $\varphi(t) \triangleq t - 1 - \log t$, for which $\varphi'(t) = 1 - \frac{1}{t} \geq 0$ on $[1, \infty)$. Hence $\varphi(t_k)$ is eventually nondecreasing, and because $t_k \to \infty$,

$$\varphi(t_k) = t_k - 1 - \log t_k \longrightarrow \infty.$$

Plugging this into Eq. (30) shows that $\mathrm{KL}(\rho_k \| \pi) \to \infty$ and is eventually nondecreasing. Therefore, for any fixed $\lambda > 0$,

$$\mathcal{C}_k(\lambda) \triangleq \frac{\mathrm{KL}(\rho_k \| \pi) + \log(1/\delta)}{\lambda}$$

is eventually nondecreasing and satisfies $\mathcal{C}_k(\lambda) \to \infty$ as $k \to \infty$. Since the remaining terms in Eq. (12) are finite for fixed $\lambda$ and $\mathbb{E}_{\theta \sim \rho_k}[L_S(\theta)] \in [0, C]$, the right-hand side of Eq. (12) diverges along such a trajectory. Consequently, the certificate worsens for sufficiently large $k$ and becomes vacuous. $\square$

## A.2. Proof of Lemma 3.2

*Proof of Lemma 3.2.* Let $p_k$ and $q_k$ be the marginals induced by the Markov kernels $(P_i)_{i \geq 1}$ and $(Q_i)_{i \geq 1}$:

$$p_k(d\theta_k) = \int P_k(d\theta_k|\theta_{k-1})\, p_{k-1}(d\theta_{k-1}), \qquad q_k(d\theta_k) = \int Q_k(d\theta_k|\theta_{k-1})\, q_{k-1}(d\theta_{k-1}).$$

Introduce the path measures on $(\theta_0, \ldots, \theta_k)$:

$$p_{0:k}(d\theta_{0:k}) \triangleq p_0(d\theta_0) \prod_{i=1}^{k} P_i(d\theta_i|\theta_{i-1}),$$

$$q_{0:k}(d\theta_{0:k}) \triangleq q_0(d\theta_0) \prod_{i=1}^{k} Q_i(d\theta_i|\theta_{i-1}),$$

so that $p_k$ and $q_k$ are the $\theta_k$-marginals of $p_{0:k}$ and $q_{0:k}$, respectively.

**Data processing (marginalization) reduces KL.** Let $\Pi_k$ be the projection map $\Pi_k(\theta_{0:k}) = \theta_k$. By the data processing inequality (i.e., contraction of KL under measurable maps),

$$\mathrm{KL}(p_k\|q_k) = \mathrm{KL}(\Pi_{k\#}p_{0:k} \,\|\, \Pi_{k\#}q_{0:k}) \;\leq\; \mathrm{KL}(p_{0:k}\|q_{0:k}), \tag{31}$$

where $\Pi_{k\#}$ denotes the pushforward measure.

**Chain rule for KL on Markov path measures.** If $p_{0:k} \not\ll q_{0:k}$, then $\mathrm{KL}(p_{0:k}\|q_{0:k}) = +\infty$ and the claim is trivial, so assume $p_{0:k} \ll q_{0:k}$. Using the factorization of the Radon–Nikodym derivative,

$$\frac{dp_{0:k}}{dq_{0:k}}(\theta_{0:k}) = \frac{dp_0}{dq_0}(\theta_0) \prod_{i=1}^{k} \frac{dP_i(\cdot|\theta_{i-1})}{dQ_i(\cdot|\theta_{i-1})}(\theta_i),$$

we obtain

$$\begin{aligned}
\mathrm{KL}(p_{0:k}\|q_{0:k}) &= \mathbb{E}_{p_{0:k}}\left[\log \frac{dp_{0:k}}{dq_{0:k}}(\theta_{0:k})\right] \\
&= \mathbb{E}_{p_{0:k}}\left[\log \frac{dp_0}{dq_0}(\theta_0)\right] + \sum_{i=1}^{k} \mathbb{E}_{p_{0:k}}\left[\log \frac{dP_i(\cdot|\theta_{i-1})}{dQ_i(\cdot|\theta_{i-1})}(\theta_i)\right] \\
&= \mathrm{KL}(p_0\|q_0) + \sum_{i=1}^{k} \mathbb{E}_{\theta_{i-1}\sim p_{i-1}}\left[\mathrm{KL}(P_i(\cdot|\theta_{i-1})\|Q_i(\cdot|\theta_{i-1}))\right].
\end{aligned} \tag{32}$$

The last equality follows from the tower property under $p_{0:k}$ and the definition of conditional KL: conditioning on $\theta_{i-1}$, the inner expectation becomes the KL between the conditional laws $P_i(\cdot|\theta_{i-1})$ and $Q_i(\cdot|\theta_{i-1})$.

**Combine.** Combining Eq. (31) with Eq. (32) yields

$$\mathrm{KL}(p_k\|q_k) \leq \mathrm{KL}(p_0\|q_0) + \sum_{i=0}^{k-1} \mathbb{E}_{\theta_i\sim p_i}\left[\mathrm{KL}\left(P_{i+1}(\cdot|\theta_i)\|Q_{i+1}(\cdot|\theta_i)\right)\right],$$

which is exactly Eq. (13). The final statement follows by setting $p_0 = q_0$. $\qquad\square$

## A.3. Proof of Theorem 3.3

*Proof of Theorem 3.3.* We prove each claim in turn.

**Step 1: Marginal KL is controlled by the sum of expected transition KLs.** By Lemma 3.2, for any $k \geq 1$,

$$\mathrm{KL}(p_k\|q_k) \leq \mathrm{KL}(p_0\|q_0) + \sum_{i=0}^{k-1} \mathbb{E}_{\theta_i \sim p_i}\left[\mathrm{KL}\left(P_{i+1}(\cdot|\theta_i)\|Q_{i+1}(\cdot|\theta_i)\right)\right].$$

Under the assumption $p_0 = q_0$, the initial term vanishes, yielding Eq. (20).

**Step 2: Closed-form transition KL for matched-mean isotropic Gaussians.** Fix $i \geq 1$ and condition on $\theta_{i-1}$. By Eq. (14)–(15), $P_i(\cdot|\theta_{i-1})$ and $Q_i(\cdot|\theta_{i-1})$ are Gaussians with identical mean $\theta_{i-1} - \eta_i g_i(\theta_{i-1})$ and isotropic covariances $\sigma_i^2 \eta_i^2 I$ and $\sigma_\lambda^2 \eta^2 I$, respectively. Therefore, by the standard Gaussian KL formula,

$$\mathrm{KL}(P_i\|Q_i) = \frac{d}{2}\left(\frac{\sigma_i^2 \eta_i^2}{\sigma_\lambda^2 \eta^2} - 1 - \log\frac{\sigma_i^2 \eta_i^2}{\sigma_\lambda^2 \eta^2}\right).$$

With $\eta_i = \alpha_i \eta$ from Eq. (17), this becomes

$$\mathrm{KL}(P_i\|Q_i) = \frac{d}{2}\left(r_i - 1 - \log r_i\right), \qquad r_i \triangleq \frac{\sigma_i^2 \alpha_i^2}{\sigma_\lambda^2},$$

which is Eq. (21).

**Step 3: The TVA gate enforces $r_i \leq 1$ (up to a negligible $\varepsilon$ effect).** Recall the gate Eq. (18):

$$\alpha_i = \min\left\{\frac{\sigma_i + \varepsilon}{\sigma_\lambda}, \; \frac{\sigma_\lambda}{\sigma_i + \varepsilon}\right\}.$$

We show $\sigma_i \alpha_i \leq \sigma_\lambda$ for all $\sigma_i \geq 0$, which implies $r_i \leq 1$.

*Case 1:* $\sigma_i + \varepsilon \leq \sigma_\lambda$. Then $\alpha_i = \frac{\sigma_i+\varepsilon}{\sigma_\lambda}$, so

$$\sigma_i \alpha_i = \frac{\sigma_i(\sigma_i + \varepsilon)}{\sigma_\lambda} \leq \frac{(\sigma_i + \varepsilon)^2}{\sigma_\lambda} \leq \frac{\sigma_\lambda^2}{\sigma_\lambda} = \sigma_\lambda,$$

where we used $\sigma_i \leq \sigma_i + \varepsilon$ and $\sigma_i + \varepsilon \leq \sigma_\lambda$.

*Case 2:* $\sigma_i + \varepsilon \geq \sigma_\lambda$. Then $\alpha_i = \frac{\sigma_\lambda}{\sigma_i+\varepsilon}$, so

$$\sigma_i \alpha_i = \sigma_i \frac{\sigma_\lambda}{\sigma_i + \varepsilon} \leq \sigma_\lambda \quad \text{since} \quad \frac{\sigma_i}{\sigma_i + \varepsilon} \leq 1.$$

Thus $\sigma_i \alpha_i \leq \sigma_\lambda$ always, and therefore

$$r_i = \frac{\sigma_i^2 \alpha_i^2}{\sigma_\lambda^2} \leq 1.$$

**Step 4: Uniform boundedness of $\mathrm{KL}(P_i\|Q_i)$ in the variance-inflation regime.** Assume $\sigma_i \geq \sigma_\lambda$. Then $\sigma_i + \varepsilon \geq \sigma_\lambda$ and hence $\alpha_i = \frac{\sigma_\lambda}{\sigma_i+\varepsilon}$, yielding

$$r_i = \frac{\sigma_i^2}{(\sigma_i + \varepsilon)^2} = \left(\frac{\sigma_i}{\sigma_i + \varepsilon}\right)^2.$$

The function $h(\sigma) = \frac{\sigma}{\sigma+\varepsilon}$ is increasing on $(0, \infty)$, so for $\sigma_i \geq \sigma_\lambda$,

$$r_i \in \left[\left(\frac{\sigma_\lambda}{\sigma_\lambda + \varepsilon}\right)^2, \; 1\right),$$

which is Eq. (22). Define $r_\star \triangleq \left(\frac{\sigma_\lambda}{\sigma_\lambda+\varepsilon}\right)^2 \in (0, 1)$ and $\varphi(r) \triangleq r - 1 - \log r$ for $r > 0$. On $(0, 1]$ we have $\varphi'(r) = 1 - \frac{1}{r} \leq 0$, hence $\varphi$ is decreasing on $(0, 1]$. Since $r_i \in [r_\star, 1)$, it follows that $\varphi(r_i) \leq \varphi(r_\star)$ and therefore

$$\mathrm{KL}(P_i\|Q_i) = \frac{d}{2}\varphi(r_i) \leq \frac{d}{2}\varphi(r_\star) = \frac{d}{2}\left(r_\star - 1 - \log r_\star\right),$$

a finite constant independent of $\sigma_i$ hence uniformly bounded over the regime $\sigma_i \geq \sigma_\lambda$. Moreover, as $\sigma_i \to \infty$ with $\varepsilon > 0$ fixed, $r_i \to 1$ and thus $\mathrm{KL}(P_i\|Q_i) \to 0$, showing explicitly that the per-step transition KL does not blow up as $\sigma_i$ becomes large.

**Step 5: Conclude.** Step 1 gives the marginal bound Eq. (20). Steps 2–4 show that TVA enforces $r_i \leq 1$ and yields a uniform upper bound on each transition KL term in the variance-inflation regime $\sigma_i \geq \sigma_\lambda$. Consequently, TVA prevents variance-driven explosion of the transition-KL contributions when $\sigma_i$ becomes large, and therefore controls an upper bound on marginal KL drift through Eq. (20). □

### A.4. Overall Pseudocode of TVA Algorithm

---

**Algorithm 1** Transition-variance alignment.

---

**Require:** Base learning rate $\eta$, Initial model parameter $\hat{\theta}_0$, Reference variance $\sigma_\lambda$, Dimensionality of parameter $d$, Time-series dataset $\{S_1, \ldots, S_K\}$
$m_0 \leftarrow \hat{\theta}_0, V_0 \leftarrow \mathbf{0}, \hat{g}_0 \leftarrow 0, \epsilon = 1 \times 10^{-8}$
**for** $k = 1$ **to** $K$ **do**
   $g_k \leftarrow \text{Adam}(\widehat{\nabla}L_{S_k}(m_k))$

   $\bar{g}_k \leftarrow 1/k \sum_d (g_k + \hat{g}_{k-1})$
   $\sigma_k^2 \leftarrow 1/d \, (g_k - \bar{g}_k)^2$
   $\alpha_k \leftarrow \exp\left(-\left|\log(\sigma_k + \varepsilon) - \log \sigma_\lambda\right|\right)$

   $m_k \leftarrow m_{k-1} - \alpha_k \eta \, g_k(m_{k-1})$
   $V_k \leftarrow V_{k-1} + \sigma_k^2 (\alpha_k \eta)^2 I$

   $\hat{g}_k \leftarrow (g_k + \hat{g}_{k-1})$
**end for**

---

While the theoretical development in Section 3 is presented using a generic stochastic optimization framework, our experiments employ Adam optimization in practice. Accordingly, Algorithm 1 summarizes the overall TVA procedure integrated with Adam-based updates. At each training step, TVA first computes the stochastic gradient and the corresponding Adam-preconditioned update, since the actual parameter update is determined after optimizer preconditioning. TVA then estimates the transition-variance scale from the deviation of the preconditioned update relative to its running mean using a mean-field approximation. Here, the running mean is introduced solely to construct the centered update required by the variance definition, rather than to perform EMA-style smoothing of the transition-variance itself. Based on the discrepancy between the estimation transition-variance and the target scale, TVA computes an adaptive gate factor that modulates the global step size. The resulting gated update stabilizes parameter transition dynamics while remaining fully compatible with standard adaptive optimizers such as Adam.

## B. Implementation Details

### B.1. Dataset Description

We evaluated the performance of TVA on eight widely used time-series forecasting datasets (Zhou et al., 2021; Wu et al., 2021) for multivariate time-series forecasting. The detailed statistics for each dataset are summarized in Table 5.

**ETT series** contain oil temperature and load measurements from power transformers, collected between July 2016 and July 2018. ETTh1 and ETTh2 were sampled at hourly intervals, while ETTm1 and ETTm2 were recorded at every 15-minute intervals. All four sub-datasets share seven common variables (channels).

*Table 5.* Statistics of benchmark datasets.

| Dataset | ETTh1 | ETTh2 | ETTm1 | ETTm2 | Electricity | Solar-Energy | Traffic | Weather |
|---------|-------|-------|-------|-------|-------------|--------------|---------|---------|
| Channels | 7 | 7 | 7 | 7 | 321 | 137 | 862 | 21 |
| Timesteps | 17,420 | 17,420 | 69,680 | 69,680 | 26,304 | 52,560 | 17,544 | 52,696 |
| Frequency | 1 hour | 1 hour | 15 min | 15 min | 1 hour | 10 min | 1 hour | 10 min |

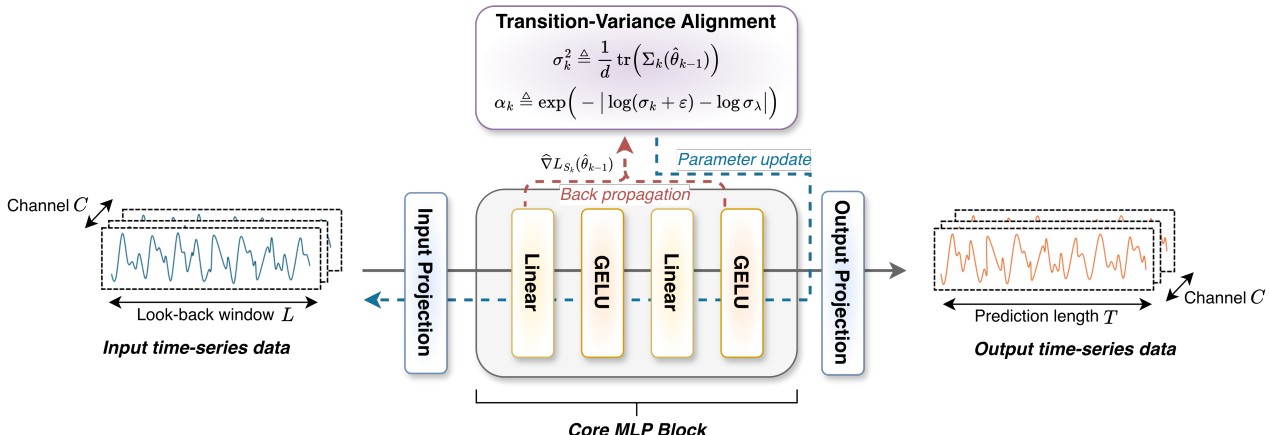

*Figure 5.* Overall model architecture and the integration of TVA.

**Electricity (ECL)** records the hourly electricity consumption of 321 customers from 2016 to 2019.

**Solar-Energy (Solar)** contains solar power generation data collected from 137 photovoltaic (PV) power plants in 2006, sampled at 10-minute intervals.

**Traffic** includes road occupancy measurements collected from 862 sensors deployed on freeways in the San Francisco Bay Area between 2015 and 2016, recorded at hourly intervals.

**Weather (WTH)** consists of meteorological observations collected at 10-minute intervals throughout 2020, comprising 21 indicators including temperature, humidity, and related atmospheric variables.

### B.2. MLP Backbone and Integration of TVA

Figure 5 illustrates the architecture of the MLP-based backbone used in our experiments, along with the integration point of the proposed TVA algorithm. Following prior work (Lin et al., 2024a), we designed the MLP backbone as follows:

**Input Projection Layer.** The input tensor has shape $(B, C, L)$, where $B$ is the batch size, $C$ is the number of input variables (channels), and $L$ is the look-back window, denoting the length of historical observations provided as input. This tensor is first permuted to $(B, L, C)$ to better align the temporal dimension for processing. A linear transformation is then applied along the temporal axis to project the sequence of length $L$ into a latent dimension $d$, enabling the model to capture temporal dependencies across channels more effectively.

**Core MLP Block.** The projected input is passed through two fully connected layers with GELU activation in between. This MLP block serves as the main feature extractor, transforming temporal representations into a richer latent space suitable for forecasting.

**Output Projection Layer.** The hidden representation is then linearly projected from the latent dimension $d$ to the prediction length $T$, representing the number of future time steps to forecast. Optionally, a dropout layer may be inserted before the projection. The resulting tensor is finally permuted back to the shape $(B, T, C)$ to match the format of the ground truth labels.

**Integration of TVA.** TVA operates independently of the model. It estimates parameter-wise transition-variance by comparing the current model parameters with their previous values and uses this information to interpolate between them. This mechanism adaptively suppresses inter-parameter correlation without modifying the model structure or introducing additional trainable parameters. The effect is a more stable training trajectory and improved generalization.

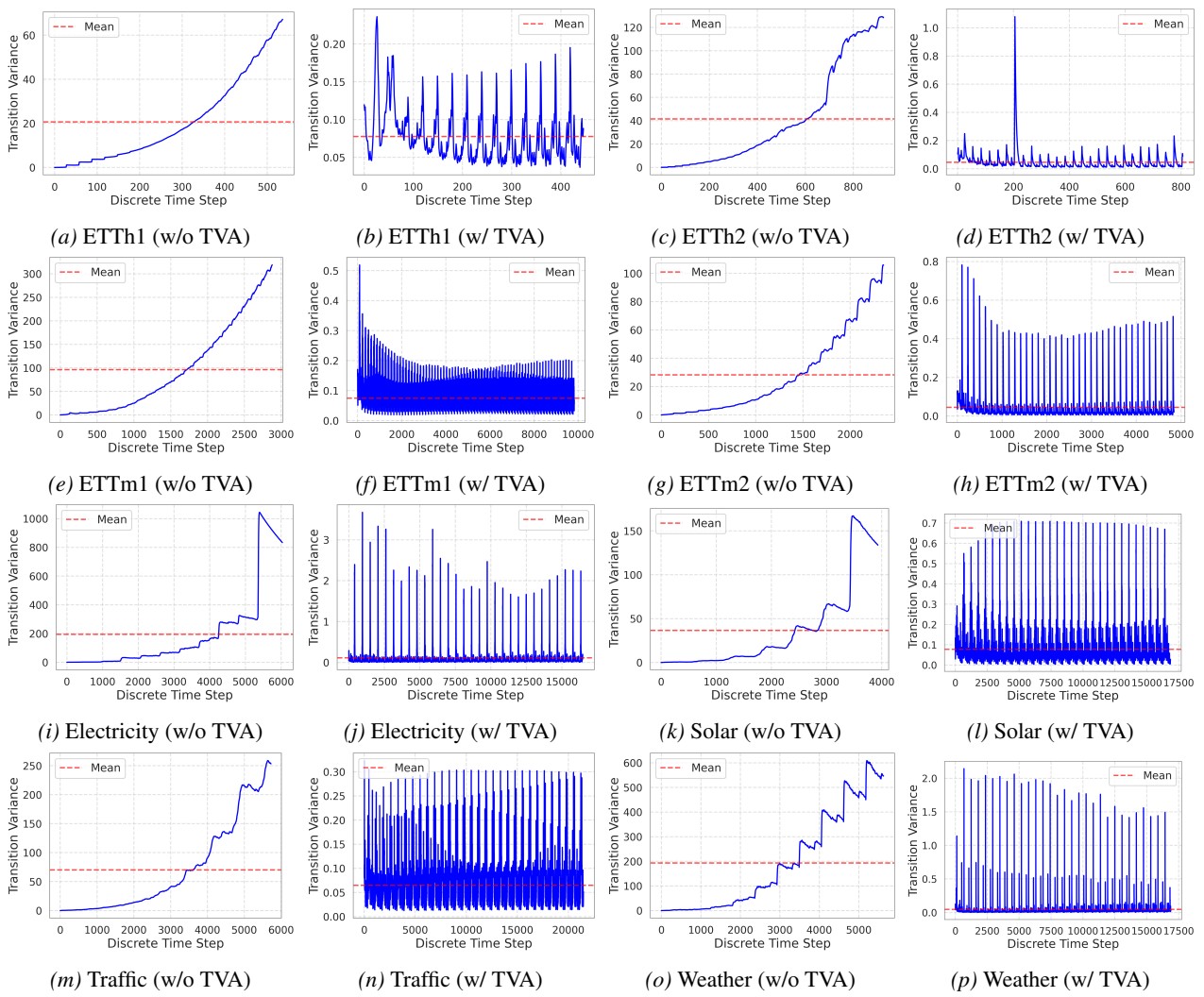

*Figure 6.* Transition-variance dynamics across training steps in various datasets, with and without TVA. The look-back window $L$ is fixed to 96, and the prediction length $T$ is fixed to 720.

## C. Additional Results of TVA

### C.1. Transition-variance Dynamics with and without TVA

To further support the transition-variance analysis in Section 4.3, we examined the evolution of parameter transition-variance over discrete training steps across multiple benchmark datasets. Figure 6 compares the empirical transition-variance under two configurations: (1) training without TVA (w/o TVA) and (2) training with TVA (w/ TVA). Each pair of plots corresponds to a specific dataset and illustrates how the transition-variance evolves throughout training.

In the absence of TVA, the transition-variance progressively increased across all datasets, which corresponds to unstable parameter updates. This behavior is consistent with the hypothesis discussed in the Methodology section: temporally ordered time-series samples induce correlated gradients, which amplify variance accumulation during optimization. In contrast, when TVA was applied, the transition-variance remained bounded and fluctuated around a stable mean, as indicated by the red dashed lines in Figure 6. This stabilization arises from TVA's adaptive step-size modulation, which dynamically scales parameter updates according to the discrepancy between empirical and target transition-variance levels, thereby constraining transition-variance growth.

This effect is further corroborated by the kernel density estimation plots in Figure 7. For vanilla models, the transition-variance distributions corresponding to the initial and final training stages exhibited a significant shift, reflecting progressive drift in update uncertainty and non-stationary optimization dynamics. In contrast, when TVA was integrated, the transition-

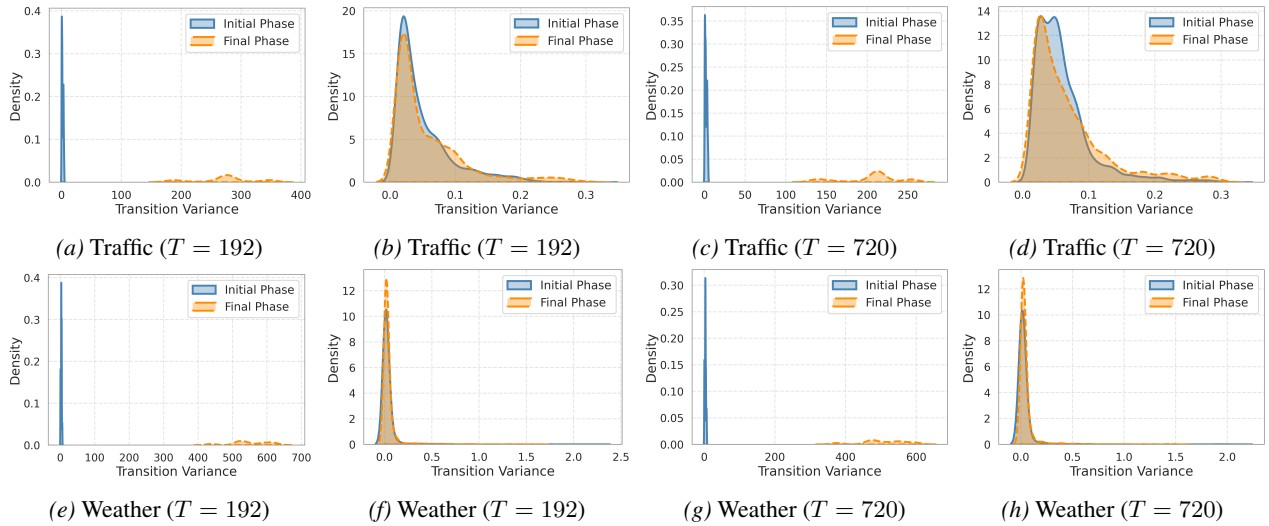

*Figure 7.* Kernel density estimation plots of transition-variance distributions computed from the first 20% (*Initial Phase*) and last 20% (*Final Phase*) of the sequence for the Traffic and Weather datasets. The look-back window $L$ is fixed to 96, and the prediction length $T$ is set to 192 or 720. Columns 1 and 3 correspond to the MLP baseline without TVA, while columns 2 and 4 show results with TVA.

variance distributions remained sharply concentrated near zero, forming narrow and high peaks, presenting predictable parameter updates. Moreover, the distributions from the initial and final training phases largely overlapped, as shown in the second and fourth columns of Figure 7. This phase-wise consistency demonstrates that cumulative drift in update uncertainty is effectively suppressed over the course of training.

By enforcing a stable transition-variance regime, TVA prevents the induced parameter distribution from progressively deviating during optimization, aligning with the theoretical motivation of controlling marginal KL divergence through bounded transition-wise variability. Overall, these results support the core idea underlying TVA: explicit regulation of transition-variance curbs excessive parameter correlation induced by temporally correlated gradients, stabilizes parameter dynamics, and ultimately improves generalization for robust multivariate time-series forecasting.

### C.2. Comparison with Standard Learning-Rate Schedulers

Since TVA operates by modulating the effective step size, we examine whether its effect can be reproduced by standard learning-rate scheduling. To this end, we compare MLP+TVA with two widely used schedulers applied to the MLP backbone: cosine annealing and ReduceLROnPlateau. All experiments followed the same protocol as the main experiments. The reported results are MSE values averaged over all prediction lengths $T \in \{96, 192, 336, 720\}$.

As shown in Table 6, TVA achieved lower MSE than the standard schedulers on all datasets. While standard adaptive

*Table 6.* Comparison between TVA and standard learning-rate schedulers. Results are reported as MSE averaged over all prediction lengths $T \in \{96, 192, 336, 720\}$. **Bold** indicates the lowest MSE in each row, and red values with upward arrows(↑) denote the relative MSE increase compared to MLP+TVA.

|  | MLP+TVA | MLP+cosine annealing | MLP+ReduceLROnPlateau |
|---|---|---|---|
| ETTh1 | **0.438** | 0.459 (↑4.79%) | 0.458 (↑4.57%) |
| ETTh2 | **0.371** | 0.373 (↑0.54%) | 0.374 (↑0.81%) |
| ETTm1 | **0.389** | 0.391 (↑0.51%) | 0.390 (↑0.26%) |
| ETTm2 | **0.278** | 0.291 (↑4.68%) | 0.291 (↑4.68%) |
| Electricity | **0.195** | 0.196 (↑0.51%) | 0.213 (↑9.23%) |
| Solar | **0.234** | 0.275 (↑17.52%) | 0.262 (↑11.97%) |
| Traffic | **0.452** | 0.473 (↑4.65%) | 0.511 (↑13.05%) |
| Weather | **0.260** | 0.268 (↑3.08%) | 0.279 (↑7.31%) |

*Table 7.* Comparison of MLP+TVA using the mean-field scale approximation and the diagonal covariance variant. Results are reported as MSE averaged over all prediction lengths $T \in \{96, 192, 336, 720\}$. ECL and WTH denote the Electricity and Weather datasets, respectively. Shaded rows indicate the default setting used in the paper.

|  | ETTh1 | ETTh2 | ETTm1 | ETTm2 | ECL | Solar | Traffic | WTH |
|---|---|---|---|---|---|---|---|---|
| MLP+TVA(mean-field scale, default) | 0.438 | 0.371 | 0.389 | 0.278 | 0.195 | 0.234 | 0.452 | 0.260 |
| MLP+TVA(diagonal covariance) | 0.439 | 0.373 | 0.391 | 0.279 | 0.196 | 0.224 | 0.453 | 0.261 |

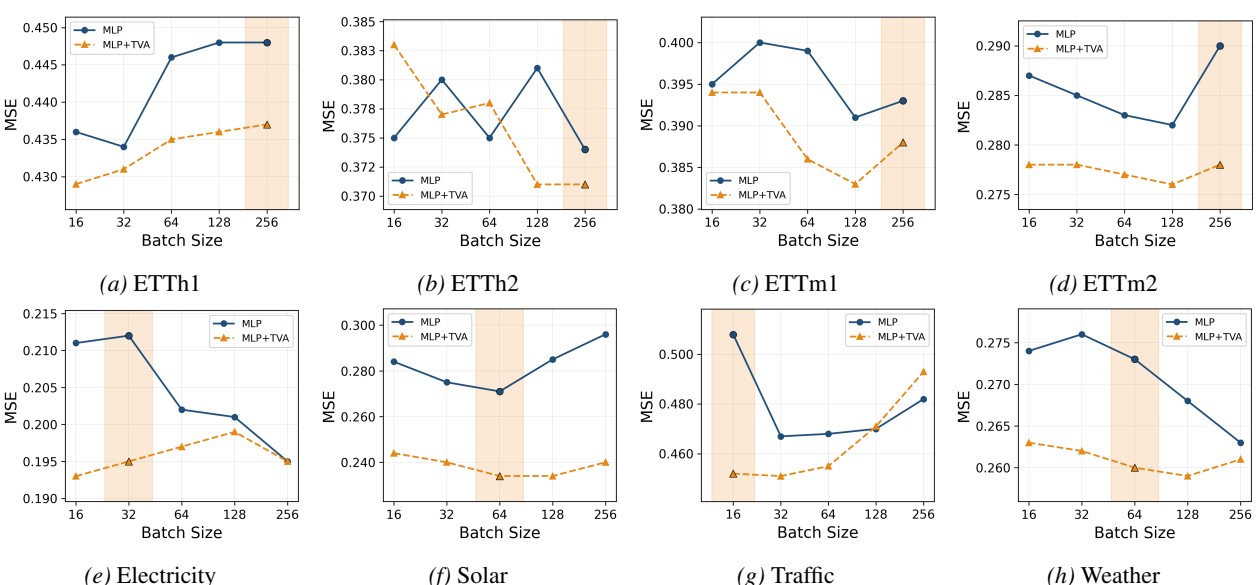

*Figure 8.* Batch size sensitivity analysis across various datasets. MSE is averaged over prediction lengths $T \in \{96, 192, 336, 720\}$ with a fixed look-back window $L = 96$. Shaded regions denote the default batch size configuration used throughout the paper.

learning-rate schedulers typically adjusted the step size based on external criteria, such as epoch-wise schedules or validation plateaus, TVA adjusted the effective step size at each update using the transition-variance signal $\sigma_k^2$ defined in Section 3.2. Furthermore, as described in Section 3.6, TVA modifies the optimization trajectory through the gate $\alpha_k$ defined in Section 3.5, thereby stabilizing the induced parameter distribution. Therefore, the advantage of TVA does not arise merely from step-size damping, but from a parameter dynamics-aware control mechanism.

### C.3. Effect of Covariance Approximation

As described in Section 3.2, we used a mean-field scale approximation instead of directly estimating the full gradient-noise covariance matrix. This choice is also consistent with prior SDE-based studies (Li et al., 2019; 2021; Malladi et al., 2022), which point out that directly handling a large dense covariance matrix is impractical in high-dimensional parameter spaces.

To examine whether the mean-field scale approximation is sufficiently effective in practice, we further considered a diagonal covariance variant, which provides a richer structure than an isotropic scalar approximation. Table 7 compares the performance of MLP with TVA using the mean-field scale approximation and its diagonal covariance variant. The reported results are MSE values averaged over all prediction lengths $T \in \{96, 192, 336, 720\}$.

On the Solar dataset, using the diagonal covariance improved the MSE from 0.234 to 0.224. However, on the remaining datasets, the mean-field scale approximation achieved nearly identical or slightly better performance than the diagonal covariance variant. These results indicate that the mean-field scale approximation is sufficiently strong and efficient. Rather than being a merely convenient simplification, it represents a practical and deliberate design choice that balances predictive performance, estimation stability, and computational efficiency.

*Table 8.* Sensitivity analysis of MLP with and without TVA under different window overlap settings. Results are reported as MSE averaged over all prediction lengths $T \in \{96, 192, 336, 720\}$. Gain (%) denotes the relative improvement of MLP+TVA over the vanilla MLP. Shaded rows indicate the default overlap setting used in the paper.

| Dataset | Overlap rate (stride) | MLP | MLP+TVA | Gain (%) |
|---|---|---|---|---|
| Solar | 99% ($s = 1$) | 0.277 | 0.234 | +15.5 |
|  | 75% ($s = 24$) | 0.319 | 0.284 | +11.0 |
|  | 50% ($s = 48$) | 0.438 | 0.425 | +3.0 |
|  | 0% ($s = 96$) | 0.470 | 0.415 | +11.7 |
| Traffic | 99% ($s = 1$) | 0.508 | 0.452 | +11.0 |
|  | 75% ($s = 24$) | 0.878 | 0.862 | +1.8 |
|  | 50% ($s = 48$) | 0.859 | 0.838 | +2.4 |
|  | 0% ($s = 96$) | 0.919 | 0.540 | +41.2 |

## C.4. Sensitivity Analysis of Batch Size

To analyze the sensitivity of TVA to batch size, we varied the batch size over $\{16, 32, 64, 128, 256\}$ and compared the performance of MLP and MLP with TVA. Figure 8 reports the MSE averaged over all prediction lengths for each dataset. Overall, MLP with TVA achieved lower MSE than the vanilla MLP across a wide range of batch sizes, showing relatively stable performance under batch size variations. In particular, on ETTh1, ETTm2, Electricity, Solar and Weather, the model with TVA consistently recorded lower errors across most batch size settings. For Solar and Weather, the vanilla MLP exhibited relatively noticeable performance fluctuations as the batch size changed, whereas MLP+TVA maintained lower MSE with only mild variations.

On some datasets, the gap between the two methods was small at certain batch sizes, and the vanilla MLP showed competitive performance in some large batch regimes, such as Traffic. Nevertheless, TVA generally exhibited more stable performance across diverse batch size settings and achieved lower MSE than the vanilla MLP on most datasets. This indicates that TVA is not a method that depends on a specific batch size; rather, it maintains stable performance by regulating the uncertainty of parameter transitions even under gradient-noise regimes that vary with batch size.

## C.5. Sensitivity Analysis of Window Overlap

To analyze the sensitivity of TVA to window overlap, we controlled the overlap rate by varying the stride in sliding-window sampling. Specifically, we used strides $s \in \{1, 24, 48, 96\}$, corresponding to approximately $99\%, 75\%, 50\%$, and $0\%$ overlap, respectively, and compared the performance of MLP and MLP with TVA on the Solar and Traffic datasets. Here, the $99\%$ overlap setting corresponds to stride $s = 1$, which is the default setting used in the paper. Table 8 reports the MSE averaged over all prediction lengths for each overlap setting together with the relative improvement achieved by TVA.

As shown in Table 8, MLP with TVA consistently achieved lower MSE than the vanilla MLP across all overlap settings on both Solar and Traffic. Relative improvements ranged from +3.0% to +15.5% on Solar and from +1.8% to +41.2% on Traffic, demonstrating that TVA remains effective under diverse overlap regimes. These results indicate that the effectiveness of TVA does not depend on a specific overlap configuration. Rather, TVA maintains robust performance under varying temporal redundancy and gradient-noise regimes induced by different window overlap rates.

## C.6. Visualization of Prediction Results

To qualitatively evaluate the impact of TVA, we visualized forecasting results on four representative benchmark datasets: Electricity, Weather, Solar, and Traffic. Figure 9 compares the prediction performance of the MLP backbone with and without TVA for prediction lengths $T \in \{336, 720\}$ under a fixed look-back window of $L = 96$. The first and third columns correspond to the baseline models without TVA (w/o TVA), while the second and fourth columns display the results with TVA (w/ TVA) applied.

The visualizations revealed that the vanilla baseline frequently suffered from pronounced forecasting distortions, including phase shifts and amplitude damping, particularly as the prediction horizon increased. These failure modes typically arise when the model fails to maintain stable parameter dynamics under strong temporal dependencies, leading to inflation of the transition-variance and degraded generalization.

In contrast, TVA effectively mitigates these issues by adaptively regulating update uncertainty at the parameter level. By stabilizing the induced parameter distribution and controlling the transition-KL surrogate, TVA promotes accurate temporal alignment and preserves the structural scale of fluctuations across all evaluated datasets. These qualitative results demonstrate that TVA enhances forecasting reliability by aligning internal learning dynamics with the complex temporal dependencies inherent in real-world time-series data.

### C.7. Comparative Analysis with the Latest MLP Models

To evaluate the competitiveness of the proposed method, we conducted a comparative analysis against recent state-of-the-art MLP-based forecasting models, namely SparseTSF (Lin et al., 2025) and SOFTS (Han et al., 2024). In this experiment, SparseTSF refers to the SparseTSF/MLP variant. Table 9 reports the MSE and MAE results, comparing a vanilla MLP augmented with TVA against these specialized architectures. The results in Table 9 showed that a vanilla MLP integrated with TVA achieved performance comparable to or even exceeded, that of recent MLP-based forecasting models, despite not relying on architectural modifications or model-specific refinements. In particular, on the Solar dataset, MLP with TVA achieved an average MSE of 0.234, substantially outperforming SparseTSF/MLP (0.288) and SOFTS (0.301). The TVA-augmented MLP also achieved the best results on ETTh2, ETTm2, and Weather. These findings suggest that parameter decorrelation during training is as critical as intricate model-specific designs for enhancing generalization performance.

### C.8. Full Results of Main Table

Table 10 presents the complete multivariate forecasting results on all benchmark datasets. All experiments were conducted with a fixed look-back window of $L = 96$ and prediction lengths $T \in \{96, 192, 336, 720\}$, covering a wide range of architectures, including MLP, Transformer-based and TCN-based backbones. The results showed that TVA consistently improved forecasting accuracy over different prediction horizons and frequently achieved state-of-the-art performance relative to the corresponding baseline models in most experimental settings. This consistent improvement under varying prediction lengths and datasets supports the conclusion that TVA constitutes a robust and scalable framework, effectively stabilizing parameter dynamics to deliver reliable high-precision forecasting regardless of backbone choice or task complexity.

### C.9. Full Results of Different Normalization Techniques

Table 11 presents comprehensive experimental results for combining TVA with instance normalization techniques—RevIN (Kim et al., 2022), DishTS (Fan et al., 2023), and SAN (Liu et al., 2023)—across multiple model architectures (iTransformer (Liu et al., 2024), DLinear (Zeng et al., 2023), and MLP) and datasets. All experiments were conducted with a fixed look-back window of $L = 96$. The results consistently demonstrated that the integration of TVA with instance normalization improved forecasting accuracy across most configurations. These findings provide empirical evidence for the complementary role of TVA. While instance normalization addresses input-level distribution shifts, TVA mitigates inter-parameter correlation, which is a source of generalization error not handled by input normalization, without introducing additional model complexity or requiring architectural modifications.

### C.10. Full Results of STD Techniques

Table 12 presents the full comparison of different STD techniques for time-series forecasting, evaluated with and without the integration of the TVA method. Results are reported in terms of both MSE and MAE across all prediction lengths. Consistent with earlier findings, TVA enhanced the forecasting accuracy of STD-based models across multiple datasets, highlighting its compatibility with diverse decomposition strategies. Notably, even a vanilla Linear model augmented with TVA often matched or outperformed more complex STD-based architectures. This underscores the strong standalone effectiveness of the TVA method in capturing essential temporal dependencies through parameter decorrelation.

### C.11. Robustness and Reproducibility

To evaluate the robustness and reproducibility of the proposed method, we conducted additional experiments using five different random seeds: $\{2026, 2027, 2028, 2029, 2030\}$. As summarized in Table 13, the MLP model with TVA exhibits remarkably low variance across all seeds. Across all datasets and forecast horizons, the standard deviations for both MSE and MAE consistently remained below 0.003 and were often exactly 0.000, indicating minimal performance fluctuation due to randomness. These results confirm the stability and reliability of the proposed approach. In particular, the low variance underscores that the observed improvements do not arise from favorable random conditions but instead stem from the

intrinsic generalizability of the method. This robustness strengthens the empirical validity of our findings and highlights the practical suitability of the proposed approach for deployment in real-world time-series forecasting scenarios.

## D. Limitations

Although TVA demonstrates robust performance across a wide range of experimental settings, it currently relies on a user-specified target transition-variance. While the sensitivity analysis indicates that TVA is relatively insensitive to the precise choice of this parameter within a broad range, manual specification may limit practicality when deploying the method across heterogeneous datasets or in scenarios with limited tuning budgets. An important direction for future work is to develop an adaptive strategy for estimating or updating the target transition-variance during training. Enabling data-driven adaptation of the target variance would further improve the usability and scalability of TVA in real-world forecasting applications.

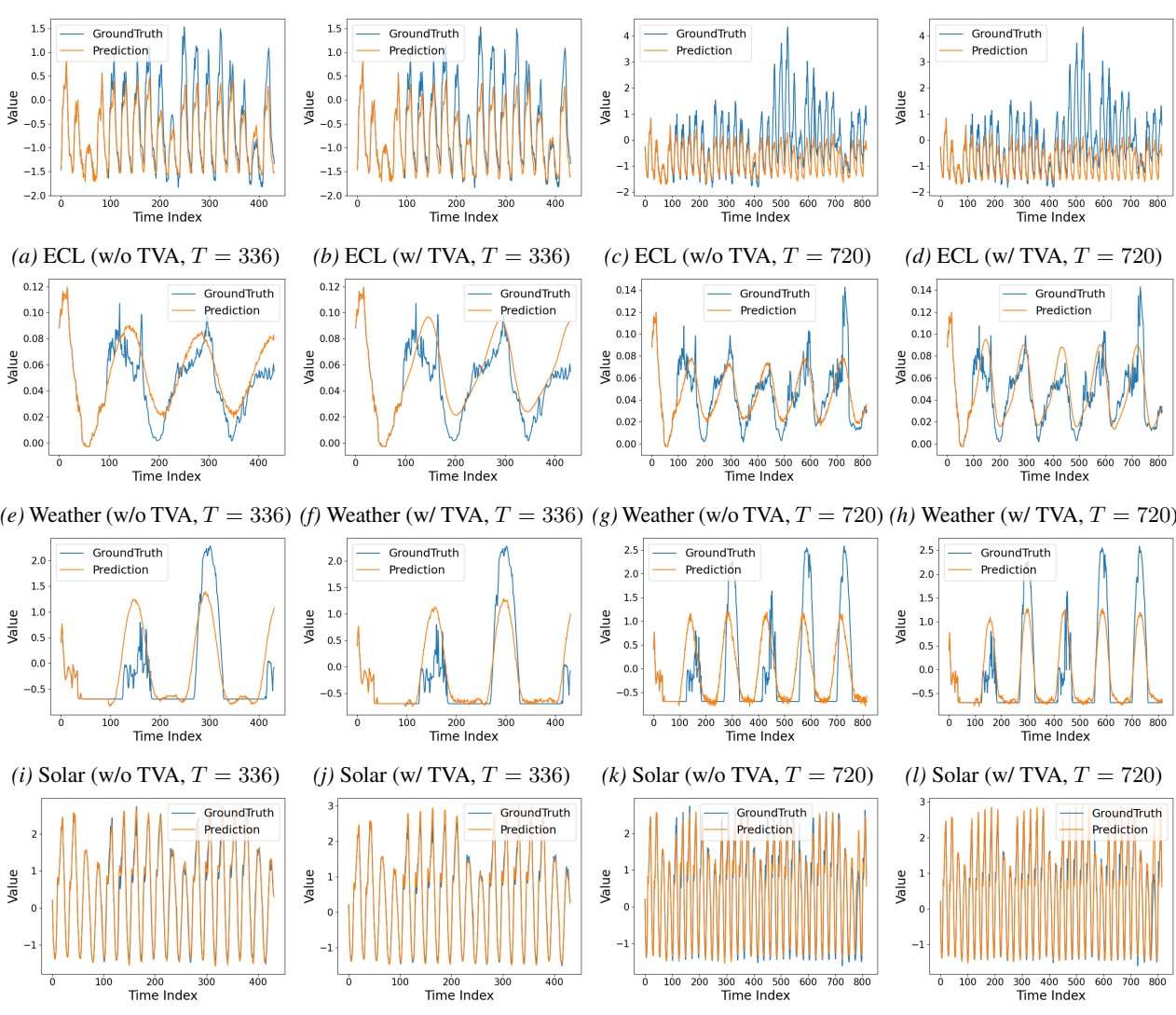

*(a)* ECL (w/o TVA, $T = 336$)   *(b)* ECL (w/ TVA, $T = 336$)   *(c)* ECL (w/o TVA, $T = 720$)   *(d)* ECL (w/ TVA, $T = 720$)

*(e)* Weather (w/o TVA, $T = 336$) *(f)* Weather (w/ TVA, $T = 336$) *(g)* Weather (w/o TVA, $T = 720$) *(h)* Weather (w/ TVA, $T = 720$)

*(i)* Solar (w/o TVA, $T = 336$)   *(j)* Solar (w/ TVA, $T = 336$)   *(k)* Solar (w/o TVA, $T = 720$)   *(l)* Solar (w/ TVA, $T = 720$)

*(m)* Traffic (w/o TVA, $T = 336$)  *(n)* Traffic (w/ TVA, $T = 336$)  *(o)* Traffic (w/o TVA, $T = 720$)  *(p)* Traffic (w/ TVA, $T = 720$)

*Figure 9.* Visualization of forecasting results for prediction lengths $T \in \{336, 720\}$ with a fixed look-back window $L = 96$. The blue and orange lines denote the ground truth and predicted values, respectively. Columns 1 and 3 show the MLP baseline (w/o TVA), while columns 2 and 4 show the MLP with TVA (w/ TVA) results. Rows from top to bottom represent ECL, Weather, Solar, and Traffic. ECL denotes Electricity.

*Table 9.* Performance comparison between MLP with TVA and the latest MLP architectures. The look-back horizon was fixed at 96. Avg indicates the average performance across all prediction lengths. SparseTSF denotes SparseTSF/MLP. The lower MSE and MAE values are highlighted in **bold**.

| Model | | MLP | | MLP+TVA | | SparseTSF | | SOFTS | |
|---|---|---|---|---|---|---|---|---|---|
| Metric | | MSE | MAE | MSE | MAE | MSE | MAE | MSE | MAE |
| | 96 | 0.386 | 0.398 | **0.372** | 0.394 | 0.376 | **0.390** | 0.380 | 0.403 |
| | 192 | 0.438 | 0.427 | 0.428 | 0.426 | **0.425** | **0.420** | 0.429 | 0.431 |
| ETTh1 | 336 | 0.480 | 0.446 | 0.472 | 0.442 | **0.466** | **0.438** | 0.466 | 0.449 |
| | 720 | 0.487 | 0.468 | 0.478 | 0.467 | 0.479 | 0.467 | **0.463** | **0.467** |
| | Avg | 0.448 | 0.435 | 0.438 | 0.432 | 0.436 | **0.429** | **0.435** | 0.437 |
| | 96 | 0.293 | 0.344 | **0.293** | **0.340** | 0.314 | 0.358 | 0.298 | 0.35 |
| | 192 | 0.370 | 0.391 | **0.370** | **0.390** | 0.391 | 0.405 | 0.382 | 0.400 |
| ETTh2 | 336 | 0.416 | 0.428 | **0.409** | **0.423** | 0.428 | 0.436 | 0.423 | 0.434 |
| | 720 | 0.418 | 0.438 | **0.412** | **0.434** | 0.427 | 0.444 | 0.426 | 0.445 |
| | Avg | 0.374 | 0.400 | **0.371** | **0.397** | 0.390 | 0.411 | 0.382 | 0.407 |
| | 96 | 0.339 | 0.363 | **0.317** | **0.357** | 0.344 | 0.376 | 0.341 | 0.377 |
| | 192 | 0.366 | 0.385 | **0.365** | **0.382** | 0.385 | 0.396 | 0.384 | 0.402 |
| ETTm1 | 336 | **0.401** | **0.404** | 0.403 | 0.407 | 0.416 | 0.415 | 0.432 | 0.429 |
| | 720 | **0.465** | **0.444** | 0.470 | 0.446 | 0.478 | 0.448 | 0.490 | 0.468 |
| | Avg | 0.393 | 0.399 | **0.389** | **0.398** | 0.406 | 0.409 | 0.411 | 0.419 |
| | 96 | 0.191 | 0.276 | **0.175** | **0.259** | 0.187 | 0.268 | 0.192 | 0.278 |
| | 192 | 0.254 | 0.314 | **0.242** | **0.303** | 0.250 | 0.308 | 0.251 | 0.312 |
| ETTm2 | 336 | 0.313 | 0.350 | **0.299** | **0.338** | 0.308 | 0.345 | 0.333 | 0.366 |
| | 720 | 0.402 | 0.400 | **0.397** | **0.395** | 0.406 | 0.401 | 0.434 | 0.419 |
| | Avg | 0.290 | 0.335 | **0.278** | **0.324** | 0.288 | 0.330 | 0.302 | 0.344 |
| | 96 | 0.196 | 0.276 | 0.162 | 0.255 | 0.207 | 0.282 | **0.152** | **0.244** |
| | 192 | 0.183 | 0.267 | 0.172 | 0.263 | 0.216 | 0.299 | **0.166** | **0.257** |
| ECL | 336 | 0.218 | 0.298 | 0.192 | 0.284 | 0.226 | 0.308 | **0.186** | **0.279** |
| | 720 | 0.274 | 0.354 | 0.253 | 0.337 | 0.266 | 0.342 | **0.245** | **0.332** |
| | Avg | 0.218 | 0.299 | 0.195 | 0.285 | 0.229 | 0.308 | **0.187** | **0.278** |
| | 96 | 0.247 | 0.332 | **0.217** | 0.298 | 0.265 | 0.342 | 0.268 | **0.292** |
| | 192 | 0.270 | 0.339 | **0.237** | 0.318 | 0.290 | 0.358 | 0.302 | **0.316** |
| Solar | 336 | 0.282 | 0.357 | **0.238** | **0.314** | 0.292 | 0.356 | 0.318 | 0.327 |
| | 720 | 0.308 | 0.379 | **0.244** | **0.308** | 0.306 | 0.361 | 0.315 | 0.321 |
| | Avg | 0.277 | 0.352 | **0.234** | **0.310** | 0.288 | 0.354 | 0.301 | 0.314 |
| | 96 | 0.449 | 0.289 | 0.418 | 0.287 | 0.596 | 0.359 | **0.391** | **0.267** |
| | 192 | 0.511 | 0.331 | 0.441 | 0.294 | 0.570 | 0.344 | **0.417** | **0.278** |
| Traffic | 336 | 0.521 | 0.334 | 0.458 | 0.301 | 0.585 | 0.358 | **0.433** | **0.286** |
| | 720 | 0.551 | 0.354 | 0.491 | 0.320 | 0.645 | 0.375 | **0.463** | **0.302** |
| | Avg | 0.508 | 0.327 | 0.452 | 0.300 | 0.599 | 0.359 | **0.426** | **0.283** |
| | 96 | 0.197 | 0.236 | **0.182** | **0.220** | 0.206 | 0.266 | 0.204 | 0.255 |
| | 192 | 0.243 | 0.272 | **0.225** | **0.258** | 0.264 | 0.306 | 0.254 | 0.289 |
| WTH | 336 | 0.292 | 0.307 | **0.280** | **0.297** | 0.314 | 0.340 | 0.309 | 0.328 |
| | 720 | 0.360 | 0.351 | **0.354** | **0.346** | 0.381 | 0.381 | 0.379 | 0.373 |
| | Avg | 0.273 | 0.291 | **0.260** | **0.280** | 0.291 | 0.323 | 0.286 | 0.311 |

*Table 10.* Multivariate time-series forecasting results with prediction length $T \in \{96, 192, 336, 720\}$. Avg indicates the average performance across all prediction lengths. The best and second-best results are highlighted in **bold** and underlined, respectively.

| Model | | MLP | | MLP+TVA | | iTransformer | | PatchTST | | MSGNet | | Crossformer | | TimesNet | |
|---|---|---|---|---|---|---|---|---|---|---|---|---|---|---|---|
| Metric | | MSE | MAE | MSE | MAE | MSE | MAE | MSE | MAE | MSE | MAE | MSE | MAE | MSE | MAE |
| ETTh1 | 96 | 0.386 | 0.398 | **0.372** | **0.394** | 0.386 | 0.402 | 0.383 | 0.399 | 0.392 | 0.414 | 0.432 | 0.450 | 0.408 | 0.420 |
| | 192 | 0.438 | 0.427 | **0.428** | **0.426** | 0.441 | 0.434 | 0.437 | 0.429 | 0.449 | 0.453 | 0.522 | 0.515 | 0.448 | 0.445 |
| | 336 | 0.480 | 0.446 | **0.472** | **0.442** | 0.480 | 0.451 | 0.487 | 0.452 | 0.536 | 0.503 | 0.718 | 0.646 | 0.499 | 0.472 |
| | 720 | 0.487 | 0.468 | **0.478** | **0.467** | 0.489 | 0.479 | 0.499 | 0.476 | 0.497 | 0.490 | 0.646 | 0.604 | 0.515 | 0.496 |
| | Avg | 0.448 | 0.435 | **0.438** | **0.432** | 0.449 | 0.442 | 0.451 | 0.439 | 0.469 | 0.465 | 0.580 | 0.554 | 0.468 | 0.458 |
| ETTh2 | 96 | **0.293** | 0.344 | **0.293** | **0.340** | 0.301 | 0.350 | 0.318 | 0.359 | 0.311 | 0.363 | 0.644 | 0.550 | 0.326 | 0.370 |
| | 192 | **0.370** | 0.391 | **0.370** | **0.390** | 0.380 | 0.399 | 0.391 | 0.404 | 0.415 | 0.425 | 0.629 | 0.577 | 0.408 | 0.421 |
| | 336 | 0.416 | 0.428 | **0.409** | **0.423** | 0.417 | 0.431 | 0.433 | 0.436 | 0.418 | 0.436 | 0.801 | 0.653 | 0.441 | 0.445 |
| | 720 | 0.418 | 0.438 | **0.412** | **0.434** | 0.419 | 0.442 | 0.437 | 0.451 | 0.427 | 0.446 | 1.145 | 0.789 | 0.448 | 0.458 |
| | Avg | 0.374 | 0.400 | **0.371** | **0.397** | 0.379 | 0.406 | 0.395 | 0.412 | 0.393 | 0.418 | 0.805 | 0.642 | 0.406 | 0.423 |
| ETTm1 | 96 | 0.339 | 0.363 | **0.317** | **0.357** | 0.351 | 0.379 | 0.328 | 0.362 | 0.333 | 0.374 | 0.444 | 0.472 | 0.330 | 0.377 |
| | 192 | 0.366 | 0.385 | **0.365** | **0.382** | 0.387 | 0.396 | 0.370 | 0.384 | 0.399 | 0.407 | 0.462 | 0.480 | 0.402 | 0.408 |
| | 336 | **0.401** | **0.404** | 0.403 | 0.407 | 0.425 | 0.420 | 0.403 | 0.405 | 0.462 | 0.442 | 0.524 | 0.530 | 0.422 | 0.422 |
| | 720 | 0.465 | 0.444 | 0.470 | 0.446 | 0.490 | 0.456 | **0.460** | **0.438** | 0.508 | 0.464 | 0.576 | 0.562 | 0.484 | 0.457 |
| | Avg | 0.393 | 0.399 | **0.389** | 0.398 | 0.413 | 0.413 | 0.390 | **0.397** | 0.425 | 0.422 | 0.502 | 0.511 | 0.409 | 0.416 |
| ETTm2 | 96 | 0.191 | 0.276 | **0.175** | **0.259** | 0.186 | 0.271 | 0.182 | 0.268 | 0.183 | 0.265 | 0.328 | 0.408 | 0.191 | 0.271 |
| | 192 | 0.254 | 0.314 | **0.242** | **0.303** | 0.256 | 0.317 | 0.248 | 0.310 | 0.248 | 0.305 | 0.324 | 0.400 | 0.259 | 0.313 |
| | 336 | 0.313 | 0.350 | **0.299** | **0.338** | 0.317 | 0.354 | 0.313 | 0.351 | 0.315 | 0.347 | 0.499 | 0.497 | 0.320 | 0.348 |
| | 720 | 0.402 | 0.400 | **0.397** | **0.395** | 0.414 | 0.407 | 0.410 | 0.407 | 0.417 | 0.406 | 0.777 | 0.642 | 0.414 | 0.402 |
| | Avg | 0.290 | 0.335 | **0.278** | **0.324** | 0.293 | 0.337 | 0.288 | 0.334 | 0.291 | 0.331 | 0.482 | 0.487 | 0.296 | 0.333 |
| Electricity | 96 | 0.196 | 0.276 | 0.162 | 0.255 | **0.148** | **0.241** | 0.238 | 0.328 | 0.321 | 0.400 | 0.226 | 0.333 | 0.198 | 0.301 |
| | 192 | 0.183 | 0.267 | 0.172 | 0.263 | **0.162** | **0.255** | 0.234 | 0.325 | 0.299 | 0.381 | 0.230 | 0.331 | 0.213 | 0.312 |
| | 336 | 0.218 | 0.298 | 0.192 | 0.284 | **0.176** | **0.270** | 0.255 | 0.345 | 0.302 | 0.383 | 0.308 | 0.390 | 0.223 | 0.322 |
| | 720 | 0.274 | 0.354 | **0.253** | **0.337** | 0.275 | 0.354 | 0.309 | 0.387 | 0.426 | 0.476 | 0.544 | 0.536 | 0.320 | 0.390 |
| | Avg | 0.218 | 0.299 | 0.195 | 0.285 | **0.190** | **0.280** | 0.259 | 0.346 | 0.337 | 0.410 | 0.327 | 0.398 | 0.238 | 0.331 |
| Solar | 96 | 0.247 | 0.332 | **0.217** | 0.298 | 0.272 | 0.300 | 0.304 | 0.352 | 0.860 | 0.673 | 0.234 | **0.281** | 0.274 | 0.296 |
| | 192 | 0.270 | 0.339 | **0.237** | 0.318 | 0.294 | 0.313 | 0.320 | 0.347 | 0.741 | 0.576 | 0.281 | 0.322 | 0.304 | **0.308** |
| | 336 | 0.282 | 0.357 | **0.238** | 0.314 | 0.328 | 0.332 | 0.359 | 0.356 | 0.495 | 0.445 | 0.283 | 0.316 | 0.338 | 0.324 |
| | 720 | 0.308 | 0.379 | **0.244** | **0.308** | 0.297 | 0.314 | 0.414 | 0.419 | 0.403 | 0.390 | 0.290 | 0.323 | 0.350 | 0.335 |
| | Avg | 0.277 | 0.352 | **0.234** | **0.310** | 0.298 | 0.315 | 0.349 | 0.369 | 0.625 | 0.521 | 0.272 | 0.311 | 0.316 | 0.316 |
| Traffic | 96 | 0.449 | 0.289 | 0.418 | 0.287 | **0.407** | **0.276** | 0.543 | 0.352 | 0.617 | 0.329 | 0.554 | 0.312 | 0.600 | 0.323 |
| | 192 | 0.511 | 0.331 | 0.441 | 0.294 | **0.425** | **0.283** | 0.530 | 0.342 | 0.655 | 0.349 | 0.581 | 0.333 | 0.630 | 0.336 |
| | 336 | 0.521 | 0.334 | 0.458 | 0.301 | **0.440** | **0.290** | 0.533 | 0.340 | 0.663 | 0.360 | 0.602 | 0.325 | 0.655 | 0.349 |
| | 720 | 0.551 | 0.354 | 0.491 | 0.320 | **0.469** | **0.307** | 0.608 | 0.370 | 0.690 | 0.383 | 0.656 | 0.347 | 0.695 | 0.367 |
| | Avg | 0.508 | 0.327 | 0.452 | 0.300 | **0.435** | **0.289** | 0.554 | 0.351 | 0.656 | 0.355 | 0.598 | 0.329 | 0.645 | 0.344 |
| Weather | 96 | 0.197 | 0.236 | 0.182 | **0.220** | 0.197 | 0.235 | 0.211 | 0.246 | 0.183 | 0.229 | 0.226 | 0.302 | **0.177** | 0.223 |
| | 192 | 0.243 | 0.272 | **0.225** | **0.258** | 0.242 | 0.272 | 0.252 | 0.281 | 0.229 | 0.268 | 0.363 | 0.435 | 0.237 | 0.271 |
| | 336 | 0.292 | 0.307 | **0.280** | **0.297** | 0.294 | 0.309 | 0.295 | 0.308 | 0.288 | 0.310 | 0.467 | 0.484 | 0.288 | 0.307 |
| | 720 | 0.360 | 0.351 | **0.354** | **0.346** | 0.367 | 0.355 | 0.364 | 0.352 | 0.359 | 0.353 | 0.480 | 0.504 | 0.366 | 0.358 |
| | Avg | 0.273 | 0.291 | **0.260** | **0.280** | 0.275 | 0.293 | 0.280 | 0.297 | 0.265 | 0.290 | 0.384 | 0.431 | 0.267 | 0.290 |

*Table 11.* Comparison of different normalization techniques for diverse models, with and without the TVA method. Results are reported using MSE. +TVA indicates whether TVA is applied. The x-mark denotes the normalization method alone, while the check-mark indicates its combination with TVA. ECL and WTH denote the Electricity and Weather datasets, respectively. The lower MSE values are highlighted in **bold**.

| Models | | iTransformer | | | | | | DLinear | | | | | | MLP | | | | |
|---|---|---|---|---|---|---|---|---|---|---|---|---|---|---|---|---|---|---|
| Norm. | | RevIN | | DishTS | | SAN | | RevIN | | DishTS | | SAN | | RevIN | | DishTS | | SAN |
| +TVA | | ✗ | ✓ | ✗ | ✓ | ✗ | ✓ | ✗ | ✓ | ✗ | ✓ | ✗ | ✓ | ✗ | ✓ | ✗ | ✓ | ✗ | ✓ |
| ETTh1 | 96 | 0.386 | **0.382** | 0.404 | **0.395** | 0.612 | **0.474** | 0.385 | **0.383** | **0.392** | 0.392 | 0.639 | **0.492** | 0.386 | **0.372** | **0.385** | 0.393 | 0.482 | **0.400** |
| | 192 | 0.441 | **0.434** | **0.444** | 0.445 | 0.683 | **0.568** | 0.436 | **0.435** | 0.451 | **0.446** | 0.648 | **0.459** | 0.438 | **0.428** | **0.457** | 0.458 | 0.521 | **0.455** |
| | 336 | 0.480 | **0.473** | 0.536 | **0.500** | 0.654 | **0.551** | 0.480 | **0.477** | 0.511 | **0.499** | 0.667 | **0.548** | 0.480 | **0.472** | 0.515 | **0.503** | 0.558 | **0.496** |
| | 720 | 0.489 | **0.482** | 0.551 | **0.541** | 0.678 | **0.523** | **0.484** | 0.489 | 0.553 | **0.537** | 0.649 | **0.546** | 0.487 | **0.478** | **0.578** | 0.593 | 0.616 | **0.527** |
| | Avg | 0.449 | **0.443** | 0.484 | **0.470** | 0.657 | **0.529** | **0.446** | 0.446 | 0.477 | **0.468** | 0.651 | **0.511** | 0.448 | **0.438** | 0.484 | 0.487 | 0.544 | **0.469** |
| ETTh2 | 96 | 0.301 | **0.300** | **0.806** | 0.869 | **0.321** | 0.322 | **0.289** | 0.297 | 0.380 | **0.328** | 0.318 | **0.315** | **0.293** | 0.293 | **0.296** | 0.329 | 0.309 | **0.307** |
| | 192 | 0.380 | **0.377** | **0.541** | 0.618 | **0.405** | 0.405 | **0.375** | 0.379 | 0.543 | **0.500** | 0.400 | **0.389** | **0.370** | 0.370 | **0.399** | 0.424 | 0.396 | **0.390** |
| | 336 | 0.417 | **0.415** | 0.792 | **0.666** | **0.443** | 0.444 | **0.414** | 0.417 | 0.637 | **0.614** | **0.445** | 0.455 | 0.416 | **0.409** | 0.573 | **0.460** | **0.442** | 0.444 |
| | 720 | **0.419** | 0.419 | 1.126 | **1.095** | 0.454 | **0.446** | 0.420 | **0.415** | 0.984 | **0.941** | **0.447** | 0.454 | 0.418 | **0.412** | **0.923** | 0.978 | 0.454 | **0.450** |
| | Avg | 0.379 | **0.378** | 0.816 | **0.812** | 0.406 | **0.404** | **0.374** | 0.377 | 0.636 | **0.596** | **0.403** | 0.403 | 0.374 | **0.371** | **0.548** | 0.548 | 0.400 | **0.398** |
| ETTm1 | 96 | 0.351 | **0.342** | 0.360 | **0.353** | 0.454 | **0.403** | 0.341 | **0.336** | 0.357 | **0.346** | 0.422 | **0.359** | 0.339 | **0.317** | 0.352 | **0.338** | 0.388 | **0.343** |
| | 192 | **0.387** | 0.387 | 0.393 | **0.387** | 0.467 | **0.435** | 0.385 | **0.380** | **0.386** | 0.402 | 0.428 | **0.387** | 0.366 | **0.365** | 0.396 | **0.395** | 0.413 | **0.373** |
| | 336 | 0.425 | **0.424** | 0.427 | **0.422** | 0.494 | **0.467** | 0.418 | **0.409** | **0.417** | 0.441 | 0.482 | **0.420** | **0.401** | 0.403 | 0.432 | **0.432** | 0.438 | **0.421** |
| | 720 | 0.490 | **0.487** | 0.503 | **0.498** | 0.557 | **0.548** | 0.484 | **0.473** | **0.484** | 0.522 | 0.530 | **0.495** | **0.465** | 0.470 | 0.501 | **0.501** | 0.507 | **0.484** |
| | Avg | 0.413 | **0.410** | 0.421 | **0.415** | 0.493 | **0.463** | 0.407 | **0.399** | **0.411** | 0.428 | 0.466 | **0.415** | 0.393 | **0.389** | 0.420 | **0.416** | 0.437 | **0.405** |
| ETTm2 | 96 | 0.186 | **0.183** | 0.233 | **0.219** | **0.189** | 0.189 | 0.186 | **0.177** | **0.187** | 0.197 | **0.189** | 0.189 | 0.191 | **0.175** | 0.201 | **0.198** | 0.192 | **0.189** |
| | 192 | 0.256 | **0.251** | 0.304 | **0.297** | **0.252** | 0.252 | 0.250 | **0.243** | **0.250** | 0.283 | **0.252** | 0.252 | 0.254 | **0.242** | **0.262** | 0.275 | 0.258 | **0.252** |
| | 336 | 0.317 | **0.312** | 0.430 | **0.417** | **0.311** | 0.311 | 0.310 | **0.306** | **0.310** | 0.325 | **0.312** | 0.312 | 0.313 | **0.299** | 0.336 | **0.327** | 0.346 | **0.314** |
| | 720 | 0.414 | **0.411** | 0.667 | **0.625** | 0.411 | **0.410** | 0.410 | **0.405** | 0.681 | **0.537** | **0.410** | 0.410 | 0.402 | **0.397** | **0.444** | 0.446 | 0.415 | **0.409** |
| | Avg | 0.293 | **0.289** | 0.409 | **0.390** | **0.291** | 0.291 | 0.289 | **0.283** | 0.357 | **0.336** | **0.291** | 0.291 | 0.290 | **0.278** | **0.310** | 0.312 | 0.303 | **0.291** |
| ECL | 96 | **0.148** | 0.154 | **0.144** | 0.178 | 0.153 | **0.151** | 0.197 | **0.183** | 0.239 | **0.220** | 0.195 | **0.185** | 0.196 | **0.162** | **0.208** | 0.211 | 0.172 | **0.161** |
| | 192 | **0.162** | 0.166 | **0.205** | 0.223 | 0.166 | **0.165** | 0.200 | **0.188** | 0.247 | **0.234** | 0.194 | **0.189** | 0.183 | **0.172** | **0.215** | 0.229 | 0.181 | **0.173** |
| | 336 | **0.176** | 0.183 | **0.224** | 0.251 | **0.178** | 0.229 | 0.218 | **0.206** | 0.270 | **0.249** | **0.209** | 0.249 | 0.218 | **0.192** | **0.235** | 0.293 | 0.210 | **0.190** |
| | 720 | 0.275 | **0.244** | **0.282** | 0.337 | 0.243 | **0.238** | 0.270 | **0.251** | 0.346 | 0.401 | 0.292 | **0.248** | 0.274 | **0.253** | **0.295** | 0.348 | 0.244 | **0.232** |
| | Avg | 0.190 | **0.187** | **0.214** | 0.247 | **0.185** | 0.196 | 0.221 | **0.207** | **0.276** | 0.276 | 0.223 | **0.217** | 0.218 | **0.195** | **0.238** | 0.270 | 0.202 | **0.189** |
| Solar | 96 | 0.272 | **0.245** | 0.238 | **0.229** | 0.237 | **0.228** | 0.311 | **0.235** | 0.294 | **0.234** | 0.263 | **0.218** | 0.247 | **0.217** | 0.244 | **0.224** | 0.230 | **0.220** |
| | 192 | 0.294 | **0.270** | 0.278 | **0.246** | 0.269 | **0.248** | 0.342 | **0.268** | 0.320 | **0.270** | 0.299 | **0.263** | 0.270 | **0.237** | 0.273 | **0.263** | **0.264** | 0.265 |
| | 336 | 0.328 | **0.289** | 0.283 | **0.260** | 0.281 | **0.271** | 0.373 | **0.285** | 0.346 | **0.305** | 0.324 | **0.254** | 0.282 | **0.238** | 0.285 | **0.264** | 0.270 | **0.260** |
| | 720 | **0.297** | 0.299 | 0.300 | **0.291** | 0.277 | **0.276** | 0.382 | **0.288** | 0.358 | **0.315** | 0.327 | **0.268** | 0.308 | **0.244** | 0.313 | **0.303** | 0.259 | **0.257** |
| | Avg | 0.298 | **0.276** | 0.275 | **0.256** | 0.266 | **0.256** | 0.352 | **0.269** | 0.330 | **0.281** | 0.303 | **0.251** | 0.277 | **0.234** | 0.279 | **0.264** | 0.256 | **0.250** |
| Traffic | 96 | **0.407** | 0.417 | **0.418** | 0.440 | 0.465 | **0.450** | 0.649 | **0.480** | 0.684 | **0.535** | 0.623 | **0.535** | 0.449 | **0.418** | 0.532 | **0.512** | 0.486 | **0.469** |
| | 192 | **0.425** | 0.435 | **0.434** | 0.461 | 0.473 | **0.467** | 0.606 | **0.494** | 0.639 | **0.555** | 0.590 | **0.531** | 0.511 | **0.441** | 0.534 | **0.531** | 0.492 | **0.479** |
| | 336 | **0.440** | 0.447 | **0.453** | 0.484 | 0.488 | **0.483** | 0.615 | **0.505** | 0.673 | **0.600** | 0.597 | **0.547** | 0.521 | **0.458** | 0.553 | **0.524** | 0.506 | **0.493** |
| | 720 | **0.469** | 0.480 | **0.489** | 0.510 | 0.530 | **0.520** | 0.649 | **0.535** | 0.689 | **0.648** | 0.647 | **0.586** | 0.551 | **0.491** | 0.584 | **0.574** | 0.549 | **0.529** |
| | Avg | **0.435** | 0.445 | **0.448** | 0.474 | 0.489 | **0.480** | 0.630 | **0.503** | 0.671 | **0.585** | 0.614 | **0.550** | 0.508 | **0.452** | 0.551 | **0.535** | 0.508 | **0.493** |
| WTH | 96 | 0.197 | **0.192** | 0.290 | **0.210** | 0.195 | **0.191** | 0.197 | **0.192** | 0.221 | 0.222 | **0.189** | 0.192 | 0.197 | **0.182** | **0.215** | 0.285 | **0.190** | 0.192 |
| | 192 | 0.242 | **0.236** | 0.319 | **0.282** | 0.247 | **0.238** | 0.241 | **0.237** | **0.274** | 0.286 | **0.231** | 0.234 | 0.243 | **0.225** | **0.266** | 0.297 | 0.233 | **0.231** |
| | 336 | 0.294 | **0.289** | **0.384** | 0.405 | 0.291 | **0.287** | 0.292 | **0.289** | **0.315** | 0.343 | **0.285** | 0.287 | 0.292 | **0.280** | **0.348** | 0.358 | 0.288 | **0.287** |
| | 720 | 0.367 | **0.362** | **0.466** | 0.554 | 0.367 | **0.361** | 0.363 | **0.361** | **0.384** | 0.488 | **0.356** | 0.364 | 0.360 | **0.354** | 0.421 | **0.397** | 0.358 | **0.351** |
| | Avg | 0.275 | **0.270** | 0.365 | **0.363** | 0.275 | **0.270** | 0.273 | **0.270** | **0.299** | 0.335 | **0.265** | 0.269 | 0.273 | **0.260** | **0.313** | 0.334 | 0.267 | **0.265** |

*Table 12.* Comparison of different STD techniques with and without TVA. The x-mark denotes the STD method alone, while the check-mark indicates its combination with TVA. The lower MSE and MAE values are highlighted in **bold**.

| Models +TVA | | CLinear ✗ | | CLinear ✓ | | LDLinear ✗ | | LDLinear ✓ | | DLinear ✗ | | DLinear ✓ | | Linear ✗ | | Linear ✓ | |
|---|---|---|---|---|---|---|---|---|---|---|---|---|---|---|---|---|---|
| Metric | | MSE | MAE | MSE | MAE | MSE | MAE | MSE | MAE | MSE | MAE | MSE | MAE | MSE | MAE | MSE | MAE |
| ETTh1 | 96 | **0.378** | **0.389** | 0.394 | 0.404 | 0.379 | 0.395 | **0.377** | **0.394** | 0.385 | **0.393** | **0.383** | 0.395 | 0.385 | **0.393** | **0.383** | 0.395 |
| | 192 | **0.432** | **0.419** | 0.436 | 0.426 | 0.428 | **0.422** | **0.427** | **0.422** | 0.436 | **0.423** | **0.435** | 0.426 | 0.437 | **0.423** | **0.434** | 0.425 |
| | 336 | 0.475 | **0.440** | 0.467 | 0.442 | 0.464 | **0.440** | 0.463 | 0.440 | 0.480 | **0.445** | 0.477 | 0.447 | 0.480 | 0.446 | **0.479** | 0.448 |
| | 720 | 0.492 | 0.475 | **0.471** | **0.472** | 0.469 | 0.462 | **0.465** | **0.460** | 0.484 | 0.471 | 0.489 | **0.435** | 0.484 | 0.471 | 0.489 | 0.475 |
| | Avg | 0.444 | **0.431** | **0.442** | 0.436 | 0.435 | 0.430 | **0.433** | **0.429** | 0.446 | 0.433 | 0.446 | **0.426** | **0.447** | 0.433 | **0.447** | 0.436 |
| ETTh2 | 96 | 0.305 | 0.351 | **0.302** | **0.347** | 0.296 | 0.343 | **0.287** | **0.338** | **0.289** | **0.337** | 0.297 | 0.344 | **0.290** | **0.339** | 0.303 | 0.349 |
| | 192 | 0.390 | 0.403 | **0.381** | **0.395** | 0.374 | 0.391 | **0.370** | **0.389** | **0.375** | **0.390** | 0.379 | 0.393 | **0.376** | **0.391** | 0.383 | 0.396 |
| | 336 | 0.439 | 0.438 | **0.425** | **0.430** | 0.414 | 0.425 | **0.408** | **0.422** | **0.414** | **0.425** | 0.417 | 0.426 | **0.415** | **0.426** | 0.421 | 0.429 |
| | 720 | 0.433 | 0.446 | **0.426** | **0.443** | 0.417 | 0.437 | **0.411** | **0.434** | 0.420 | 0.439 | **0.415** | **0.435** | 0.420 | 0.439 | 0.423 | 0.441 |
| | Avg | 0.392 | 0.409 | **0.384** | **0.404** | 0.375 | 0.399 | **0.369** | **0.396** | **0.374** | **0.398** | 0.377 | 0.400 | **0.375** | **0.399** | 0.382 | 0.404 |
| ETTm1 | 96 | **0.322** | **0.359** | 0.331 | 0.366 | 0.331 | 0.362 | **0.325** | **0.361** | 0.341 | 0.362 | **0.336** | **0.360** | 0.341 | 0.362 | **0.335** | **0.361** |
| | 192 | 0.379 | 0.394 | **0.369** | **0.386** | 0.373 | 0.384 | **0.366** | **0.382** | 0.385 | 0.386 | **0.380** | **0.384** | 0.384 | 0.385 | **0.379** | **0.384** |
| | 336 | 0.409 | 0.413 | **0.399** | **0.407** | 0.404 | 0.404 | **0.399** | **0.403** | 0.418 | 0.407 | **0.409** | **0.404** | 0.418 | 0.406 | **0.410** | **0.405** |
| | 720 | 0.466 | 0.443 | **0.458** | **0.441** | 0.461 | 0.439 | 0.462 | **0.439** | 0.484 | 0.444 | **0.473** | **0.440** | 0.484 | 0.443 | **0.475** | **0.441** |
| | Avg | 0.394 | 0.403 | **0.389** | **0.400** | 0.392 | 0.397 | **0.388** | **0.396** | 0.407 | 0.400 | **0.399** | **0.397** | 0.407 | 0.399 | **0.400** | **0.398** |
| ETTm2 | 96 | 0.179 | 0.262 | **0.166** | **0.250** | 0.177 | 0.261 | **0.176** | **0.259** | 0.186 | 0.271 | **0.177** | **0.260** | 0.196 | 0.282 | **0.177** | **0.260** |
| | 192 | 0.268 | 0.326 | **0.233** | **0.294** | 0.244 | 0.305 | **0.241** | **0.300** | 0.250 | 0.309 | **0.243** | **0.302** | 0.259 | 0.319 | **0.243** | **0.303** |
| | 336 | 0.326 | 0.361 | **0.293** | **0.333** | 0.315 | 0.351 | **0.302** | **0.339** | 0.310 | 0.347 | **0.306** | **0.342** | 0.318 | 0.355 | **0.308** | **0.344** |
| | 720 | 0.409 | 0.402 | **0.395** | **0.391** | 0.412 | 0.406 | **0.401** | **0.396** | 0.410 | 0.402 | **0.405** | **0.398** | 0.418 | 0.408 | **0.405** | **0.398** |
| | Avg | 0.295 | 0.338 | **0.272** | **0.317** | 0.287 | 0.331 | **0.280** | **0.323** | 0.289 | 0.332 | **0.283** | **0.325** | 0.298 | 0.341 | **0.283** | **0.326** |
| Electricity | 96 | 0.150 | 0.247 | **0.145** | **0.240** | 0.166 | **0.257** | **0.161** | 0.257 | 0.197 | 0.274 | **0.183** | **0.268** | 0.200 | 0.276 | **0.183** | **0.268** |
| | 192 | 0.164 | 0.257 | **0.159** | **0.252** | 0.175 | **0.265** | **0.173** | 0.266 | 0.200 | 0.279 | **0.188** | **0.273** | 0.200 | 0.279 | **0.188** | **0.273** |
| | 336 | 0.185 | 0.282 | **0.180** | **0.275** | 0.193 | **0.282** | **0.190** | 0.283 | 0.218 | 0.298 | **0.206** | **0.290** | 0.212 | 0.292 | **0.205** | **0.290** |
| | 720 | 0.242 | 0.330 | **0.222** | **0.311** | 0.232 | **0.313** | **0.230** | 0.315 | 0.270 | 0.347 | **0.251** | **0.327** | 0.266 | 0.341 | **0.249** | **0.325** |
| | Avg | 0.185 | 0.279 | **0.177** | **0.270** | 0.191 | **0.279** | **0.189** | 0.280 | 0.221 | 0.300 | **0.207** | **0.289** | 0.219 | 0.297 | **0.206** | **0.289** |
| Solar | 96 | 0.256 | 0.283 | **0.203** | **0.267** | 0.269 | 0.300 | **0.241** | **0.289** | 0.311 | 0.402 | **0.235** | **0.336** | 0.288 | 0.375 | **0.230** | **0.330** |
| | 192 | 0.294 | 0.301 | **0.242** | **0.291** | 0.293 | 0.315 | **0.253** | **0.300** | 0.342 | 0.421 | **0.268** | **0.365** | 0.319 | 0.396 | **0.259** | **0.355** |
| | 336 | 0.343 | 0.325 | **0.269** | **0.306** | 0.296 | 0.314 | **0.272** | **0.301** | 0.373 | 0.435 | **0.285** | **0.371** | 0.374 | 0.429 | **0.289** | **0.372** |
| | 720 | 0.356 | 0.333 | **0.284** | **0.321** | 0.295 | 0.310 | **0.285** | **0.309** | 0.382 | 0.436 | **0.288** | **0.376** | 0.391 | 0.441 | **0.273** | **0.358** |
| | Avg | 0.312 | 0.311 | **0.249** | **0.296** | 0.288 | 0.310 | **0.263** | **0.300** | 0.352 | 0.423 | **0.269** | **0.362** | 0.343 | 0.410 | **0.263** | **0.354** |
| Traffic | 96 | 0.483 | 0.323 | **0.468** | **0.321** | **0.423** | **0.276** | 0.434 | 0.290 | 0.649 | 0.385 | **0.480** | **0.333** | 0.648 | 0.385 | **0.480** | **0.332** |
| | 192 | 0.464 | **0.300** | **0.463** | 0.304 | 0.437 | **0.282** | **0.435** | 0.288 | 0.606 | 0.363 | **0.494** | **0.333** | 0.605 | 0.364 | **0.492** | **0.332** |
| | 336 | 0.478 | **0.309** | **0.473** | 0.310 | **0.447** | **0.287** | 0.450 | 0.294 | 0.615 | 0.366 | **0.505** | **0.334** | 0.614 | 0.366 | **0.505** | **0.334** |
| | 720 | 0.509 | 0.331 | **0.503** | **0.327** | 0.480 | **0.310** | **0.479** | 0.311 | 0.649 | 0.389 | **0.535** | **0.352** | 0.649 | 0.387 | **0.533** | **0.351** |
| | Avg | 0.483 | 0.316 | **0.477** | **0.315** | 0.447 | **0.289** | 0.449 | 0.296 | 0.630 | 0.376 | **0.503** | **0.338** | 0.629 | 0.375 | **0.503** | **0.337** |
| Weather | 96 | 0.171 | **0.216** | **0.170** | 0.217 | 0.192 | 0.230 | **0.187** | **0.226** | 0.197 | 0.235 | **0.192** | **0.230** | 0.195 | 0.234 | **0.192** | **0.230** |
| | 192 | 0.228 | 0.264 | **0.218** | **0.255** | **0.228** | 0.263 | 0.229 | **0.263** | 0.241 | 0.271 | **0.237** | **0.266** | 0.240 | 0.270 | **0.237** | **0.266** |
| | 336 | 0.280 | 0.301 | **0.272** | **0.294** | **0.280** | 0.299 | 0.281 | 0.300 | 0.292 | 0.306 | **0.289** | **0.303** | 0.291 | 0.306 | **0.289** | **0.303** |
| | 720 | 0.353 | 0.348 | **0.347** | **0.343** | **0.348** | **0.343** | 0.350 | 0.345 | 0.363 | 0.353 | **0.361** | **0.350** | 0.363 | 0.353 | **0.361** | **0.350** |
| | Avg | 0.258 | 0.282 | **0.252** | **0.277** | 0.262 | **0.284** | **0.261** | **0.284** | 0.273 | 0.291 | **0.270** | **0.287** | 0.272 | 0.291 | **0.270** | **0.287** |

*Table 13.* Performance of the TVA-enhanced MLP model evaluated under different random seeds {2026, 2027, 2028, 2029, 2030}. The look-back horizon was fixed at 96. Avg indicates the average performance across all prediction lengths, while Mean and STD denote the average and standard deviation, respectively.

| Setup | | Random Seed | | | | | | | | | | Mean | | STD | |
| | | 2026 | | 2027 | | 2028 | | 2029 | | 2030 | | | | | |
| Metric | | MSE | MAE | MSE | MAE | MSE | MAE | MSE | MAE | MSE | MAE | MSE | MAE | MSE | MAE |
|---|---|---|---|---|---|---|---|---|---|---|---|---|---|---|---|
| ETTh1 | 96 | 0.372 | 0.394 | 0.372 | 0.393 | 0.373 | 0.392 | 0.373 | 0.394 | 0.372 | 0.392 | 0.372 | 0.393 | 0.000 | 0.001 |
| | 192 | 0.428 | 0.426 | 0.428 | 0.421 | 0.429 | 0.422 | 0.428 | 0.421 | 0.429 | 0.422 | 0.428 | 0.422 | 0.001 | 0.002 |
| | 336 | 0.472 | 0.442 | 0.473 | 0.443 | 0.474 | 0.443 | 0.473 | 0.442 | 0.474 | 0.443 | 0.473 | 0.443 | 0.001 | 0.001 |
| | 720 | 0.478 | 0.467 | 0.479 | 0.468 | 0.478 | 0.467 | 0.477 | 0.466 | 0.480 | 0.467 | 0.478 | 0.467 | 0.001 | 0.001 |
| | Avg | 0.438 | 0.432 | 0.438 | 0.431 | 0.438 | 0.431 | 0.438 | 0.431 | 0.439 | 0.431 | 0.438 | 0.431 | 0.001 | 0.001 |
| ETTh2 | 96 | 0.293 | 0.340 | 0.298 | 0.344 | 0.298 | 0.343 | 0.293 | 0.341 | 0.295 | 0.343 | 0.295 | 0.342 | 0.002 | 0.002 |
| | 192 | 0.370 | 0.390 | 0.369 | 0.389 | 0.370 | 0.391 | 0.370 | 0.390 | 0.371 | 0.391 | 0.370 | 0.390 | 0.001 | 0.001 |
| | 336 | 0.409 | 0.423 | 0.409 | 0.422 | 0.409 | 0.423 | 0.408 | 0.422 | 0.409 | 0.423 | 0.409 | 0.423 | 0.000 | 0.000 |
| | 720 | 0.412 | 0.434 | 0.410 | 0.433 | 0.414 | 0.436 | 0.410 | 0.432 | 0.412 | 0.434 | 0.412 | 0.434 | 0.002 | 0.001 |
| | Avg | 0.371 | 0.397 | 0.371 | 0.397 | 0.373 | 0.398 | 0.370 | 0.396 | 0.372 | 0.398 | 0.371 | 0.397 | 0.001 | 0.001 |
| ETTm1 | 96 | 0.317 | 0.357 | 0.317 | 0.358 | 0.318 | 0.359 | 0.318 | 0.358 | 0.318 | 0.358 | 0.318 | 0.358 | 0.001 | 0.001 |
| | 192 | 0.365 | 0.382 | 0.365 | 0.382 | 0.366 | 0.383 | 0.366 | 0.383 | 0.365 | 0.382 | 0.365 | 0.383 | 0.000 | 0.001 |
| | 336 | 0.403 | 0.407 | 0.403 | 0.407 | 0.401 | 0.406 | 0.407 | 0.409 | 0.400 | 0.405 | 0.403 | 0.407 | 0.003 | 0.002 |
| | 720 | 0.470 | 0.446 | 0.465 | 0.443 | 0.466 | 0.444 | 0.466 | 0.443 | 0.466 | 0.442 | 0.467 | 0.444 | 0.002 | 0.001 |
| | Avg | 0.389 | 0.398 | 0.387 | 0.398 | 0.388 | 0.398 | 0.389 | 0.398 | 0.388 | 0.397 | 0.388 | 0.398 | 0.001 | 0.001 |
| ETTm2 | 96 | 0.175 | 0.259 | 0.175 | 0.259 | 0.175 | 0.260 | 0.175 | 0.259 | 0.175 | 0.259 | 0.175 | 0.259 | 0.000 | 0.000 |
| | 192 | 0.242 | 0.303 | 0.241 | 0.302 | 0.241 | 0.302 | 0.241 | 0.302 | 0.241 | 0.302 | 0.241 | 0.302 | 0.001 | 0.001 |
| | 336 | 0.299 | 0.338 | 0.299 | 0.338 | 0.301 | 0.340 | 0.299 | 0.338 | 0.300 | 0.339 | 0.300 | 0.339 | 0.001 | 0.001 |
| | 720 | 0.397 | 0.395 | 0.395 | 0.394 | 0.397 | 0.395 | 0.394 | 0.394 | 0.394 | 0.394 | 0.395 | 0.394 | 0.001 | 0.001 |
| | Avg | 0.278 | 0.324 | 0.278 | 0.323 | 0.278 | 0.324 | 0.277 | 0.323 | 0.278 | 0.324 | 0.278 | 0.324 | 0.000 | 0.000 |
| Electricity | 96 | 0.162 | 0.255 | 0.162 | 0.254 | 0.161 | 0.254 | 0.162 | 0.255 | 0.162 | 0.255 | 0.162 | 0.255 | 0.000 | 0.000 |
| | 192 | 0.172 | 0.263 | 0.172 | 0.263 | 0.172 | 0.263 | 0.173 | 0.264 | 0.173 | 0.264 | 0.172 | 0.263 | 0.000 | 0.000 |
| | 336 | 0.192 | 0.284 | 0.191 | 0.283 | 0.191 | 0.283 | 0.191 | 0.283 | 0.192 | 0.283 | 0.191 | 0.283 | 0.000 | 0.000 |
| | 720 | 0.253 | 0.337 | 0.256 | 0.339 | 0.252 | 0.336 | 0.254 | 0.337 | 0.251 | 0.335 | 0.253 | 0.337 | 0.002 | 0.002 |
| | Avg | 0.195 | 0.285 | 0.195 | 0.285 | 0.194 | 0.284 | 0.195 | 0.285 | 0.194 | 0.284 | 0.195 | 0.285 | 0.000 | 0.000 |
| Solar | 96 | 0.217 | 0.298 | 0.216 | 0.299 | 0.216 | 0.300 | 0.215 | 0.295 | 0.215 | 0.295 | 0.216 | 0.297 | 0.001 | 0.002 |
| | 192 | 0.237 | 0.318 | 0.239 | 0.321 | 0.239 | 0.320 | 0.239 | 0.321 | 0.239 | 0.320 | 0.239 | 0.320 | 0.001 | 0.001 |
| | 336 | 0.238 | 0.314 | 0.239 | 0.315 | 0.238 | 0.314 | 0.239 | 0.314 | 0.236 | 0.311 | 0.238 | 0.313 | 0.001 | 0.002 |
| | 720 | 0.244 | 0.308 | 0.244 | 0.307 | 0.244 | 0.306 | 0.245 | 0.308 | 0.244 | 0.305 | 0.244 | 0.307 | 0.000 | 0.001 |
| | Avg | 0.234 | 0.310 | 0.235 | 0.310 | 0.234 | 0.310 | 0.235 | 0.309 | 0.234 | 0.308 | 0.234 | 0.309 | 0.000 | 0.001 |
| Traffic | 96 | 0.418 | 0.287 | 0.417 | 0.286 | 0.419 | 0.286 | 0.418 | 0.286 | 0.417 | 0.285 | 0.418 | 0.286 | 0.001 | 0.001 |
| | 192 | 0.441 | 0.294 | 0.440 | 0.293 | 0.440 | 0.293 | 0.439 | 0.293 | 0.441 | 0.293 | 0.440 | 0.293 | 0.001 | 0.000 |
| | 336 | 0.458 | 0.301 | 0.459 | 0.302 | 0.458 | 0.302 | 0.458 | 0.301 | 0.458 | 0.302 | 0.458 | 0.301 | 0.000 | 0.000 |
| | 720 | 0.491 | 0.320 | 0.490 | 0.318 | 0.494 | 0.322 | 0.495 | 0.323 | 0.491 | 0.319 | 0.492 | 0.320 | 0.002 | 0.002 |
| | Avg | 0.452 | 0.300 | 0.452 | 0.300 | 0.453 | 0.301 | 0.453 | 0.301 | 0.452 | 0.300 | 0.452 | 0.300 | 0.001 | 0.001 |
| Weather | 96 | 0.182 | 0.220 | 0.181 | 0.220 | 0.182 | 0.220 | 0.181 | 0.220 | 0.180 | 0.219 | 0.181 | 0.220 | 0.001 | 0.001 |
| | 192 | 0.225 | 0.258 | 0.225 | 0.258 | 0.225 | 0.258 | 0.225 | 0.258 | 0.225 | 0.258 | 0.225 | 0.258 | 0.000 | 0.000 |
| | 336 | 0.280 | 0.297 | 0.279 | 0.297 | 0.279 | 0.297 | 0.279 | 0.297 | 0.279 | 0.297 | 0.279 | 0.297 | 0.000 | 0.000 |
| | 720 | 0.354 | 0.346 | 0.354 | 0.346 | 0.354 | 0.345 | 0.354 | 0.346 | 0.354 | 0.346 | 0.354 | 0.346 | 0.000 | 0.000 |
| | Avg | 0.260 | 0.280 | 0.260 | 0.280 | 0.260 | 0.280 | 0.260 | 0.280 | 0.260 | 0.280 | 0.260 | 0.280 | 0.000 | 0.000 |

