# OpenReview forum: "Parameter Decorrelation via Transition-Variance Alignment for Multivariate Time-series Forecasting"
_ICML.cc/2026/Conference — ICML 2026 regular_

### Official Review · Reviewer_k8Ad · 2026-02-14

**Soundness:** 3
**Presentation:** 3
**Significance:** 3
**Originality:** 3
**Overall Recommendation:** 4
**Confidence:** 3

**Summary:**

A key issue in multivariate time-series forecasting is that harmful parameter dependence can still arise from temporal overlap in sliding-window training, even when correlation in the data is reduced through normalization or decomposition. This effect is motivated through an optimization-as-Markov-chain and stochastic differential equation perspective, and is quantified using a scalar transition-variance signal derived from a gradient-noise-scale proxy. To address this problem, Transition-Variance Alignment (TVA) introduces a smooth, architecture-agnostic step-size gate that keeps the effective transition variance near a target level. Experiments on standard forecasting benchmarks show consistent accuracy improvements with negligible computational overhead.

**Compliance With Llm Reviewing Policy:**

Affirmed.

**Final Justification:**

Thank you for the rebuttal. It clarifies several points, especially the interaction with Adam, the target-variance search range, and the added diagonal-covariance experiment. However, my main concerns remain only partially resolved, particularly regarding the practical computation of σ, gate-factor behavior during training, and sensitivity/fairness issues relative to baselines. I appreciate the response,I give weak accept.

**Key Questions For Authors:**

How exactly is the scalar sigma computed in practice (what gradients, what variance estimate, any EMA smoothing)? How sensitive is it to batch size and window overlap?

How often does TVA downscale vs upscale the step (relative to the target)? Can you show typical trajectories of sigma and the gating factor across training?

Have you tried richer covariance surrogates (per-layer block diagonal, diagonal, or low-rank + diagonal) and do they improve robustness?

What is the tuning budget for selecting the target variance level, and is it comparable to baseline tuning? Please report search ranges and number of trials.

If experiments use Adam/AdamW, is sigma computed from raw gradients or from preconditioned updates? How does TVA interact with Adam’s adaptive scaling?

Can you report cases where TVA yields no gain or hurts, and provide an explanation (stable sigma already, weak temporal dependence, interaction with normalization)?

**Limitations:**

No. The paper would benefit from a brief limitations/broader-impact paragraph noting that TVA changes learning dynamics and could silently degrade performance under distribution shift or in high-stakes forecasting if the scalar noise proxy is unstable or misleading (especially under anisotropic gradient noise or with adaptive optimizers). The authors should clarify failure modes (mis-tuned target variance, batch/overlap sensitivity, optimizer interactions), recommend monitoring diagnostics (sigma, gate factor, drift checks), and position the method as decision support with appropriate safeguards for operational deployment.

**Strengths And Weaknesses:**

## Strengths
- TVA only gates the optimizer step size and does not require changing the forecasting backbone, making it easy to integrate.
- Connects optimization-induced dependence to a transition-variance diagnostic and motivates why controlling it can improve generalization under temporally dependent sampling.
- The intervention is lightweight and the paper reports minimal added compute.
- Experiments span multiple datasets/horizons and show compatibility with common preprocessing/normalization techniques.

## Weaknesses
- Approximating gradient-noise covariance as scalar (sigma^2 * I) may miss anisotropy/structured correlations in deep models, potentially weakening both the theoretical link and robustness.
- The target variance level is selected via a grid/greedy search; tuning budgets and fairness relative to baselines should be explicitly reported.
- Since sigma drives the gate, the exact estimator, smoothing, and sensitivity to batch size/overlap rate should be specified more clearly.
- It is unclear when TVA might hurt (already well-regularized regimes, different optimizers, very small/large batches, non-overlapping sampling).

---

> ### Author Rebuttal · Authors · 2026-03-31
>
> We thank the reviewer for the detailed review and constructive comments. We have made our best effort to address them. Below, we provide detailed responses to each question and concern.
>
> **W1. Approximating gradient-noise covariance as scalar (sigma^2 * I) may miss anisotropy/structured correlations in deep models, potentially weakening both the theoretical link and robustness:** Our covariance setting follows approximations adopted in prior work that address the impracticality of modeling a full covariance matrix in high-dimensional settings. Under this setting, we conduct our theoretical analysis. Furthermore, we provide additional experiments using a diagonal covariance matrix to support the validty of this choice. For details, please see our response to Reviewer pbzB (W3, Q3).
>
> **W2, Q4. The target variance level is selected via a grid/greedy search; tuning budgets and fairness relative to baselines should be explicitly reported. Also, please report search ranges and number of trials.:** The hyperparameter was selected via greedy search over the range {1.0, 1.5, …, 10.0}, as specified in Section 4.1. As shown in Figure 2, the proposed method remained effective across a wide range, indicating that it does not rely on extensive tuning.
>
> **W3. Since sigma drives the gate, the exact estimator, smoothing, and sensitivity to batch size/overlap rate should be specified more clearly:** As described in Section 4.1, we followed the standard training protocol for setting such as batch size and data preprocessing (e.g., overlap rate).
>
> **W4, Q6. It is unclear when TVA might hurt:** As noted in Section 4.3 and illustrated in Figure 2, we examine failure cases, where extreme choices of the transition variance scale can lead to performance degradation.
>
> ---
>
> **Q1. How exactly is the scalar sigma computed in practice? How sensitive is it to batch size and window overlap?:** As specified in pseudocode in Appendix A.4, the detailed procedure for computing the variance is provided. In addition, as noted in our response to W3, we followed the standard training protocol in our experimental setup.
>
> **Q2. How often does TVA downscale vs upscale the step (relative to the target)? Can you show typical trajectories of sigma and the gating factor across training?:** In Figure 5 (w/ TVA), the variance trajectory over time can be observed. The step-size is dynamically downscaled or upscaled based on its deviation from the target.
>
> **Q3. Have you tried richer covariance surrogates and do they improve robustness?:** As reported in W1, when using a diagonal covariance, which is closer to full covariance than the mean-field assumption, we observed comparable performance to the baseline across most datasets. These results suggest that the commonly used mean-field scale covariance remains effective. We thank the reviewer again for the insightful suggestion and will include these experiments in the Appendix to further strengthen the soundness of the paper.
>
> **Q5. If experiments use Adam/AdamW, is sigma computed from raw gradients or from preconditioned updates? How does TVA interact with Adam’s adaptive scaling?:** In our experiments, we use Adam as described in Section 4.1, and the same implementation principle applies to AdamW. In practice,  $\sigma$ is computed from the preconditioned update, i.e., the gradient scaled by Adam, rather than from the raw gradient. This choice is intentional, as TVA is designed to control the actual transition variance of parameter updates, and under Adam/AdamW the effective parameter transition is determined only after adaptive preconditioning.
>
> Therefore, TVA does not replace Adam’s coordinate-wise scaling. Instead, it operates on top of it by gating the effective global step size of the preconditioned update. This is consistent with our formulation in Section 3.1, where the update rule is introduced in an abstract stochastic-update form, and with the discussion in Section 3.2, where the SDE framework naturally extends beyond vanilla SGD to adaptive gradient methods.
>
> However, to avoid confusion, we will update the pseudocode in Appendix A.4 to more clearly describe how the TVA is combined with the Adam preconditioned update, and explicitly clarify that TVA gates the effective global step size on top of the preconditioned update, thereby improving the clarity of the presentation. We thank the reviewer for the careful comment.

---

> > ### Author Rebuttal · Reviewer_k8Ad · 2026-03-31
> >
> > The rebuttal addresses several of my questions and adds useful clarifications, especially on the interaction with Adam, the search range for the target variance, and an additional diagonal-covariance experiment. However, my concerns are only partially resolved. In particular, the rebuttal still does not clearly specify the practical computation of sigma (including smoothing/estimation details), does not directly quantify gate-factor behavior across training, and does not fully address sensitivity to batch size/window overlap or tuning-budget fairness relative to baselines.

---

> > > ### Author Response · Authors · 2026-04-05
> > >
> > > We carefully reviewed the reviewer’s comment to fully understand its intent, and we address it as follows:
> > >
> > > > does not clearly specify the practical computation of sigma (including smoothing/estimation details)
> > > >
> > >
> > > We provide further clarification on Algorithm 1 in Appendix A.4. In our method, $\sigma_k^2$ is the transition-variance introduced in Section 3.2 following prior works [a, b, c]. The variance is estimated at each training step from the centered update in an element-wise manner $1/d(g_k - \bar{g}_k)^2$, where the squared deviation is computed for each of the $d$ elements.
> > >
> > > Specifically, given the stochastic update $g_k$ and its running mean $\bar g_k$, the variance is computed as the average squared deviation. To reduce memory usage, we maintain the cumulative sum of gradients $\hat g_k$, which allows efficient computation of $\bar g_{k+1}=1/k(g_{k+1}+\hat g_k)$.
> > >
> > > We also clarify that the “smoothness” mentioned in Section 3.5 refers to the gate factor. It modulates the effective step size smoothly at $\sigma_k\approx\sigma_\lambda$ rather than applying EMA-style smoothing to $\sigma_k$. In other words, the running-average term is used only to construct the centered estimate, while the smooth behavior in Section 3.5 pertains to the response of the gate.
> > >
> > > We thank the reviewer and will include this in Appendix A.4 to improve the clarity of the practical computation.
> > >
> > > **References**
> > >
> > > [a] “Stochastic modified equations and dynamics of stochastic gradient algorithms i: Mathematical foundations.” Journal of Machine Learning Research, 2019.
> > >
> > > [b] “On the validity of modeling sgd with stochastic differential equations (sdes).” NeurIPS, 2021.
> > >
> > > [c] “On the SDEs and scaling rules for adaptive gradient algorithms.” NeurIPS, 2022.
> > >
> > > ---
> > >
> > > > does not directly quantify gate-factor behavior across training
> > > >
> > >
> > > As noted in our previous response, the behavior of the gate-factor follows Eq. (18) in Section 3.5, and can be inferred from the transition-variance trajectory in Figure 5.
> > >
> > > To provide a more direct view of the mechanism, we additionally visualize the gate-factor trajectory, which can be described as follows. In the early stages of training, as the transition-variance increases, the left-hand side of the min operation in Eq. (18) is frequently activated, causing the gate-factor to approach 1. Occasionally, for samples corresponding to the spikes observed in Figure 5, the right-hand side is activated to suppress inflation. As training progresses and the transition-variance is regulated, the right-hand side becomes more frequently activated, leading to more cases where the gate-factor takes values below 1.
> > >
> > > We thank the reviewer and will include this plot in Appendix C.1 to further clarify the underlying mechanism.
> > >
> > > ---
> > >
> > > > does not fully address sensitivity to batch size/window overlap
> > > >
> > >
> > > As discussed in our earlier response, we ensured fairness by following the standard benchmark settings for batch size and window overlap. Nevertheless, we additionally examine how performance varies with these factors. The results in the tables report the MSE averaged over prediction lengths $T \in$ {$96, 192, 336, 720$}.
> > >
> > > |  | batch size | MLP | MLP+TVA |
> > > | --- | --- | --- | --- |
> > > | Solar | 16 | 0.284 | 0.244 |
> > > |  | 32 | 0.275 | 0.240 |
> > > |  | 64 | 0.277 | 0.234 |
> > > | Traffic | 16 | 0.508 | 0.452 |
> > > |  | 32 | 0.467 | 0.451 |
> > > |  | 64 | 0.468 | 0.455 |
> > >
> > > For batch size sensitivity, TVA achieves lower MSE than the baseline across all batch sizes. The default batch sizes are 64 for Solar and 16 for Traffic, following the benchmark configuration.
> > >
> > > |  | overlap rate (stride) | MLP | MLP+TVA |
> > > | --- | --- | --- | --- |
> > > | Solar | 99% (1) | 0.277 | 0.234 |
> > > |  | 75% (24) | 0.319 | 0.284 |
> > > |  | 50% (48) | 0.438 | 0.425 |
> > > |  | 0% (96) | 0.470 | 0.415 |
> > > | Traffic | 99% (1) | 0.508 | 0.452 |
> > > |  | 75% (24) | 0.878 | 0.862 |
> > > |  | 50% (48) | 0.859 | 0.838 |
> > > |  | 0% (96) | 0.919 | 0.540 |
> > >
> > > For window overlap sensitivity, TVA achieves lower MSE than the baseline across all overlap settings. The default overlap ratio is 99%, following the benchmark setup. These results indicate that TVA is not tied to a specific batch size or window overlap setting, and remains effective across a range of configurations.
> > >
> > > We thank the reviewer for the insightful suggestion and will include these experiments in the Appendix to further improve the presentation of the paper.
> > >
> > > ---
> > >
> > > > does not fully address tuning-budget fairness relative to baselines
> > > >
> > >
> > > As noted in our earlier rebuttal, TVA does not rely on extensive tuning. As shown in Figure 2, it maintains strong performance over a wide range of $\sigma_\lambda$ values, indicating that its effectiveness stems from a robust, dynamics-aware control mechanism rather than precise tuning. Moreover, additional experiments suggested by the reviewer show that TVA exhibits low sensitivity to batch size and window overlap, further reinforcing that only minimal tuning is required overall.

---

### Official Review · Reviewer_ccPz · 2026-03-12

**Soundness:** 3
**Presentation:** 3
**Significance:** 2
**Originality:** 2
**Overall Recommendation:** 3
**Confidence:** 3

**Summary:**

This paper studies multivariate time-series forecasting from an optimization point of view. It models training as a Markov chain in parameter space and uses a scalar transition-variance signal to control the optimizer step size. The proposed TVA method is simple and can be added to different backbones.

**Compliance With Llm Reviewing Policy:**

Affirmed.

**Ethical Review Concerns:**

No ethical concerns

**Key Questions For Authors:**

Could the authors provide direct empirical evidence that TVA reduces parameter correlation or update correlation? For example, it would help to show correlation/covariance of parameter updates, cosine similarity between consecutive updates, or another direct measure of parameter dependence, with and without TVA.

**Limitations:**

ok

**Strengths And Weaknesses:**

Strengths :

- The main idea is interesting and original. Looking at forecasting through parameter dynamics, instead of only data normalization/decomposition, is a useful angle.
- The method is simple and lightweight, since it only changes the step size and does not change the backbone architecture.
- Empirically, the method is strong within the paper’s chosen baseline set. In Table 1, MLP+TVA is very competitive and often best.

Weaknesses:
- The paper claims parameter decorrelation, but the method only controls a scalar transition-variance and approximates the full covariance by \sigma_ k^2 I. The paper does not directly show that parameter correlation or update correlation is reduced.
- The PAC-Bayes part feels more like motivation than a strong justification of the method. The theory assumes bounded loss, while the paper defines loss as negative log-likelihood, and the experiments report MSE/MAE. The relation between the theorem and the actual training setup is not fully clear.
- There is also a gap between theory and experiments: the theory is written for simple parameter updates, but the experiments use Adam.
- The covariance modeling is heavily simplified. Equation (7) contains Jacobian terms, but Equation (24) later keeps only an additive isotropic variance term. This simplification needs more explanation.
- The theoretical claim is limited: Theorem 3.3 bounds each transition KL term in the large-variance regime, but Equation (20) still sums those terms over all steps. So the paper does not show that the total marginal KL stays bounded through long training.
- The empirical evaluation is not statistically strong enough in the main paper. The main tables do not report variance across runs, confidence intervals, or significance tests, and only one fixed seed is mentioned.
- Some experimental claims are too strong. Tables 2 and 3 contain several regressions, so “consistent improvements” is overstated. Also, the method is called parameter-free, but \sigma_\lambda is tuned by search.
- Finally, the SOTA comparison is incomplete. The reported results are competitive and relevant, but stronger recent baselines such as SOFTS, CycleNet, and especially TimeMixer++ are missing.

---

> ### Author Rebuttal · Authors · 2026-03-31
>
> We thank the reviewer for the careful review and constructive feedback. We have made our best effort to address them. Below, we provide detailed responses to each question and concern.
>
> **W1, Q1:** For the concern that “the method only controls a scalar transition-variance and approximates the full covariance,” please see our response to Reviewer pbzB (W3, Q3). In addition, for the concern that “the paper does not directly show the parameter or update correlation is reduced,” we refer to our response to Reviewer pbzB (W1, Q1).
>
> **W2. The relation between the PAC-Bayes theorem and the actual training setup is not fully clear:** This concern appears to stem from a confusion between the parameter distribution induced by stochastic optimization and the data likelihood used to define the training loss. Our PAC-Bayes analysis explicitly pertains to the former. In Section 3.2, we model optimization as a Markov chain in parameter space, and Theorem 3.3 shows that TVA controls variance growth by bounding the transition-KL terms in the variance-inflation regime.  Accordingly, the PAC-Bayes component serves to justify TVA as a mechanism for controlling the spread of the learned parameter distribution, rather than for directly modeling the data distribution.
>
> The data distribution is incorporated separately through the loss, which is defined in the paper as the negative log-likelihood. Moreover, under a Gaussian predictive model, the negative log-likelihood reduces to an MSE objective up to an additive constant. Hence, the use of MSE/MAE in the experiments is fully consistent with the likelihood-based formulation.
>
> **W3. The theory is written for simple parameter updates, but the experiments use Adam:** We introduce the update rule in Section 3.1 in an abstract stochastic-update form. In Section 3.2, we explain that stochastic gradient methods can be represented via SDE approximations and explicitly cite prior work on adaptive gradient algorithms [a], supporting that the same stochastic-update framework naturally extends to Adam.
>
> In our experiments, TVA is applied by scaling the step size of the optimizer, while Adam is retained as the base optimizer following standard benchmarks. Thus, TVA is derived at the level of stochastic parameter transitions, and is implemented in experiments through Adam as a standard adaptive realization of this update framework.
>
> **W4. Equation (7) contains Jacobian terms, but Equation (24) later keeps only an additive isotropic variance term:** As emphasized in our response to W1, the Jacobian term is already specialized to the isotropic-variance form in Eq. (10), as in prior works. This choice is then consistently used throughout the subsequent theoretical development and in the derivation of the practical algorithm (Section 3). Moreover, this modeling choice is empirically justified by our additional experiments in our response to Reviewer pbzB (W3, Q3).
>
> **W5. The paper does not show that the total marginal KL stays bounded through long training:** In our framework, the key quantity to control is the step-wise transition KL, as Lemma 3.2 shows that the marginal KL drift is upper-bounded by the accumulation of these transition-KL terms. Theorem 3.3 further proves that TVA uniformly bounds the contribution of each step in the variance-inflation regime, thereby preventing variance-driven explosion of these terms. This yields a meaningful and principled local generalization-control result: TVA removes the mechanism by which increasing transition variance leads to uncontrolled growth of the KL surrogate.
>
> In summary, our claim is not that TVA guarantees a globally bounded marginal KL over the entire training trajectory, but rather that it provides a theoretically grounded local control mechanism that stabilizes the growth of the marginal-KL upper bound.
>
> **W6. Lacks statistical rigor:** As shown in Appendix C.7, we conducted experiments using five different random seeds and reported both the mean and standard deviation.
>
> **W7. The phrase “consistent improvements” is overstated. Also, the method is called parameter-free, but \sigma_\lambda is tuned by search:** The term “parameter-free” in Section 4.3 (line number 412) does not refer to being hyperparameter-free; rather, it emphasizes that TVA does not modify the model architecture or introduce additional trainable parameters. However, to avoid potential ambiguity, we will remove this expression. Additionally, we will replace the term consistently with generally to better capture the overall tendency.
>
> **W8. The reported results are competitive, but stronger recent baselines such as SOFTS, CycleNet, and especially TimeMixer++ are missing:** We have already included recent methods such as SOFTS (Table 6) and CycleNet (CLinear, Table 3) in our experiments.
>
> ---
>
> **Reference**
>
> [a] Malladi, S., Lyu, K., Panigrahi, A., and Arora, S. “On the SDEs and scaling rules for adaptive gradient algorithms.” NeurIPS, 2022.

---

> > ### Author Rebuttal · Reviewer_ccPz · 2026-04-01
> >
> > Thank you for the clarification. Your rebuttal addresses an important part of my concern about the empirical comparison, statistical rigor. Besides, a second look at your tables make the practical benefit of TVA more obvious. This improves my view of the paper. I still keep a small reservation about the direct evidence for the claimed decorrelation effect, and thus I will maintain my grading.

---

> > > ### Author Response · Authors · 2026-04-05
> > >
> > > We truly appreciate the reviewer’s time and thoughtful feedback. However, we would like to reiterate that Figure 5 already shows that the transition-variance is kept bounded in practice.

---

### Official Review · Reviewer_4JqR · 2026-03-12

**Soundness:** 3
**Presentation:** 3
**Significance:** 2
**Originality:** 2
**Overall Recommendation:** 4
**Confidence:** 2

**Summary:**

This paper proposes **Parameter Decorrelation**, a training strategy designed to reduce redundancy and correlation among model parameters in time series forecasting networks. The main motivation is that many deep forecasting models contain highly correlated parameters across channels or layers, which can lead to inefficient representations and reduced generalization performance.

The proposed approach introduces a decorrelation regularization during training to encourage parameters to learn more independent representations. By reducing parameter redundancy, the method aims to improve model robustness and forecasting accuracy while keeping the overall architecture unchanged. The technique can be integrated into existing forecasting models without requiring major modifications. Experimental results across several datasets show that parameter decorrelation can improve performance compared with standard training strategies.

**Compliance With Llm Reviewing Policy:**

Affirmed.

**Final Justification:**

My concerns have been addressed.

**Key Questions For Authors:**

- How sensitive is the method to the strength of the decorrelation regularization?

- Does the benefit of parameter decorrelation persist for very large models or foundation-scale time-series models?

- Are there specific types of datasets where parameter decorrelation provides larger benefits?

- Could similar improvements be achieved through architectural design rather than training regularization?

**Limitations:**

- The approach introduces additional regularization terms that may increase training complexity.

- Improvements may depend on model size or dataset characteristics.

- The method focuses only on parameter-level decorrelation rather than representation-level disentanglement.

- The framework does not analyze the relationship between decorrelated parameters and improved forecasting interpretability.

**Strengths And Weaknesses:**

## Strengths

- **Addresses a meaningful modeling issue.** Parameter redundancy and correlation are common in deep neural networks, and explicitly encouraging decorrelated representations is a reasonable way to improve efficiency and generalization.

- **Simple and easy to integrate.** The method is implemented as a training regularization and does not require redesigning the forecasting architecture, making it broadly applicable.

- **Model-agnostic framework.** The approach can potentially be applied to many existing forecasting models, increasing its practical value.

- **Empirical improvements.** Experiments show performance gains over baseline training strategies across multiple forecasting benchmarks.

---

## Weaknesses

- **Limited methodological novelty.** Parameter decorrelation and orthogonality-based regularization have been explored extensively in prior deep learning literature.

- **Unclear impact on representation learning.** The paper does not deeply analyze how decorrelation changes the internal representations of forecasting models.

- **Potential training instability.** Strong decorrelation constraints may interfere with optimization or slow down convergence.

- **Limited theoretical analysis.** The paper mainly provides empirical evidence but lacks a deeper theoretical explanation of why decorrelation improves forecasting performance.

---

> ### Author Rebuttal · Authors · 2026-03-31
>
> We thank the reviewer for the thoughtful review and constructive comments. We have made our best effort to address them. Below, we provide responses to each question and concern.
>
> **W1. Parameter decorrelation and orthogonality-based regularization have been explored extensively in prior deep learning literature:** The line of research mentioned by the reviewer typically introduces additional loss terms to enforce orthogonality among parameters. In contrast, the key contribution of TVA lies in using transition variance as an online control signal from an SDE perspective, without introducing additional losses or explicitly enforcing orthogonality, making it fundamentally different in nature. Meanwhile, we were unable to find prior work where the suggested orthogonality-based techniques are directly applied to MTSF. Exploring the application of these techniques to MTSF appears to be an interesting direction for future work.
>
> **W2. Unclear impact on representation learning:** Interpreting our method from a representation learning perspective is an interesting direction for future work. However, as stated in Section 1, our study focuses on the over-correlated parameter issue that has been largely overlooked in existing MTSF literature.
>
> **W3. Potential training instability:** Across extensive experiments, the proposed method demonstrated performance improvements (Section 4 and Appendix C). Furthermore, training stability was validated through the sensitivity analysis in Figure 2. These results indicate that the proposed method enables stable training.
>
> **W4. Limited theoretical analysis:** Our theoretical framework is presented through Lemma 3.1, Lemma 3.2, and Theorem 3.3. Lemma 3.1 shows that variance inflation makes the PAC-Bayes complexity term non-decreasing and eventually divergent, thereby degrading the generalization bound. Lemma 3.2 shows that the marginal KL drift can be controlled via the accumulated transition-KL terms. And Theorem 3.3 shows that, in the variance-inflation regime, TVA prevents variance-driven explosion by uniformly bounding these transition-KL contributions.
>
> Therefore, our work goes beyond empirical evidence by providing a concrete theoretical mechanism that links correlation inflation to degraded generalization performance. In particular, it explains why controlling transition variance stabilizes the induced parameter distribution, mitigates overfitting, and ultimately improves forecasting performance.
>
> ---
>
> **Q1. How sensitive is the method to the strength of the decorrelation regularization?:** As shown in the sensitivity analysis in Figure 2, the proposed TVA remained effective across a wide range of settings and demonstrated strong robustness.
>
> **Q2. Does the benefit of parameter decorrelation persist for very large models or foundation-scale time-series models?:** We applied TVA to recent forecasting models and observed reliable performance improvements (Section 4 and Appendix C). Based on these results, we believe the proposed method has the potential for generalization to larger-scale and foundational-scale models.
>
> **Q3. Are there specific types of datasets where parameter decorrelation provides larger benefits?:** We observed comparable performance across all datasets, and it is difficult to attribute the gains to any specific type of dataset (Table 1 to 3). Notably, these performance suggest that transition variance inflation appears to be a common issue in existing methods, which is further illustrated in Figure 5.
>
> **Q4. Could similar improvements be achieved through architectural design rather than training regularization?:** We study the problem from a parameter dynamics perspective. Architectural design is beyond the scope of this work, but exploring similar effects along that line is an interesting direction for future research.
>
> ---
>
> **L1. The approach introduces additional regularization terms that may increase training complexity:** TVA does not introduce additional regularization terms, but instead performs scalar gating to adjust the step-size. As a result, it requires no changes to the model architecture and introduces no additional trainable parameters. Moreover, the computational overhead per batch is minimal, leading to negligible increases in overall training time (Table 4).
>
> **L2. Improvements may depend on model size or dataset characteristics:** As discussed in Q3, the proposed method was not limited to a specific model or dataset and showed performance improvements across a range of model scales and datasets (Table 4). This suggests that the effectiveness of TVA does not heavily depend on specific conditions and is broadly applicable.
>
> **L4. The framework does not analyze the relationship between decorrelated parameters and improved forecasting interpretability:** As mentioned in our response to W4, we analyzed the generalization effect theoretically through Theorem 3.3. Furthermore, we examined the corresponding effect empirically in Figure 5.

---

> > ### Author Rebuttal · Reviewer_4JqR · 2026-04-04
> >
> > My concerns have been addressed.

---

> > > ### Author Response · Authors · 2026-04-05
> > >
> > > We sincerely thank the reviewer for their time and consideration.

---

### Official Review · Reviewer_pbzB · 2026-03-19

**Soundness:** 3
**Presentation:** 3
**Significance:** 2
**Originality:** 2
**Overall Recommendation:** 4
**Confidence:** 3

**Summary:**

This paper investigates a subtle but important issue in time series forecasting: because training data are drawn from overlapping sliding windows, mini-batch samples are temporally dependent, leading to correlated stochastic gradients and consequently correlated parameter updates across optimization steps. To analyze this, the paper models stochastic optimization as a Markov chain in parameter space and views SGD through the lens of stochastic differential equations, where each update induces a Gaussian-like transition distribution and the key quantity of interest is the transition variance — the step-to-step variance attributable to gradient noise. The central empirical observation is that this transition variance often grows rather than stabilizes during training, which the authors argue is detrimental to generalization; this is further connected to a PAC-Bayes perspective, where an expanding parameter variance causes the KL-based complexity term to become prohibitively large and the generalization bound vacuous. Based on this analysis, the paper proposes Transition-Variance Alignment (TVA), a lightweight and architecture-agnostic training rule that does not modify the forecasting model itself but instead smoothly gates the learning rate at each step by estimating the current gradient noise scale and comparing it against a prescribed target, reducing the effective step size whenever the transition variance deviates from the target in either direction. Across real-world multivariate forecasting benchmarks, TVA consistently improves forecasting accuracy.

**Compliance With Llm Reviewing Policy:**

Affirmed.

**Final Justification:**

The authors' rebuttal addressed my concerns. Considering the novelty and significance of the work, I maintain my score weak accept.

**Key Questions For Authors:**

1. Could the authors provide direct empirical evidence that TVA reduces gradient or parameter correlation during training — for example, by plotting transition variance trajectories or gradient autocorrelation statistics with and without TVA?

2. Could the authors include ablation comparisons against standard adaptive learning rate schedules (e.g., cosine annealing, ReduceLROnPlateau) to isolate the specific contribution of the transition-variance alignment principle, and rule out the possibility that simple step-size damping accounts for the observed gains?

3. The theoretical motivation centers on decorrelating the full parameter update distribution, yet the algorithm controls only the scalar transition variance via an isotropic approximation. Could the authors discuss the implications of this simplification — in particular, whether TVA can still provide meaningful correction when gradient noise is highly anisotropic or directionally structured?

**Limitations:**

yes

**Strengths And Weaknesses:**

Strengths:

1. The paper is well-written and easy to follow.

2. The optimization-centric view of multivariate time series forecasting is novel and thought-provoking, and the proposed method is straightforward to implement and integrate into existing pipelines.


Weaknesses:

1. While the paper demonstrates improved forecasting accuracy empirically, it does not directly verify whether TVA actually decorrelates parameter updates as intended. Diagnostic evidence — such as measuring gradient correlation or transition variance trajectories with and without TVA — would strengthen the causal argument.

2. TVA is essentially a learning rate gating mechanism. It is unclear whether a simpler, off-the-shelf adaptive learning rate scheduler (e.g., Adam with cosine annealing or cyclical learning rates) could achieve similar gains. Without such a comparison, it is difficult to determine whether TVA's benefit stems from the transition-variance principle specifically, or merely from introducing a useful additional form of step-size damping.

3. The method is motivated as a means of mitigating parameter correlation, yet the practical algorithm replaces the full gradient-noise covariance with an isotropic scalar approximation, controlling only the magnitude of the transition variance. As a result, TVA regulates update magnitude rather than the directional correlation structure among parameters, which represents a gap between the theoretical motivation and the practical algorithm.

---

> ### Author Rebuttal · Authors · 2026-03-31
>
> We thank the reviewer for the thoughtful review and constructive feedback. We have made our best effort to address them. Below, we provide responses to each question and concern.
>
> **W1, Q1. It does not directly verify whether TVA actually decorrelates parameter updates as intended:** The transition variance trajectories are presented in Figure 5 of Appendix C.1. Without TVA (w/o TVA), the transition variance consistently increased across all datasets. In contrast, with TVA (w/ TVA), it remained bounded and fluctuated around a stable mean. Furthermore, the kernel density estimation plots in Figure 6 provide additional evidence of parameter decorrelation. These results support that TVA induces decorrelation of parameter updates as intended.
>
> **W2, Q2. TVA is essentially a learning rate gating mechanism. It is unclear whether a simpler, off-the-shelf adaptive learning rate scheduler (e.g., Adam with cosine annealing or cyclical learning rates) could achieve similar gains:** We conducted additional experiments using cosine annealing and ReduceLROnPlateau, and the results are as follows:
>
> |  | MLP+TVA(ours) | MLP+cosine annealing | MLP+ReduceLROnPlateau |
> | --- | --- | --- | --- |
> | Solar | **0.234** | 0.275 | 0.262 |
> | Traffic | **0.452** | 0.473 | 0.511 |
> | Weather | **0.260** | 0.268 | 0.279 |
>
> The reported results correspond to the MSE averaged over all prediction lengths $T \in$ {$96, 192, 336, 720$}. As shown in the results, TVA achieved lower MSE compared to cosine annealing and ReduceLROnPlateau. While standard adaptive learning rate schedulers typically adjust step size based on external criteria such as epoch-based schedules or validation plateaus, TVA adjusts the effective step size at each update using the transition-variance signal $\sigma_k^2$ defined in Section 3.2. Furthermore, as described in Section 3.6, TVA modifies the optimization trajectory through the gate $\alpha_k$ defined in Section 3.5, thereby stabilizing the induced parameter distribution. Therefore, the advantage of TVA arises not merely from step-size damping, but from a parameter dynamics-aware control mechanism. We appreciate the reviewer’s insightful suggestion and will include these experimental results in the Appendix to further improve the clarity of the paper.
>
> **W3, Q3. The method is motivated as a means of mitigating parameter correlation, yet the practical algorithm replaces the full gradient-noise covariance with an isotropic scalar approximation:** As the reviewer noted, using a full covariance can enable more flexible modeling by capturing richer dependencies. However, our mean-field scale covariance follows prior SDE-based works [a, b, c], which point out that in high-dimensional settings, handling a large dense covariance matrix is impractical. Accordingly, we explicitly state that “estimating the full matrix” is generally intractable and adopt this setting (Section 3.2).
>
> The subsequent theoretical development is consistent with this choice. Lemma 3.2 shows that controlling the step-wise transition KL leads to control over the marginal KL drift, and Theorem 3.3 demonstrates that, in the variance-inflation regime, each transition-KL contribution is uniformly bounded, thereby preventing variance-driven explosion.
>
> Nonetheless, extending TVA to control full covariance (i.e., non-isotropic covariance), as suggested by the reviewer, is an interesting direction for future work and may lead to further performance improvements. We additionally conducted experiments using a diagonal covariance, which captures richer structure than the isotropic assumption, and the results are as follows:
>
> |  | MLP + TVA (mean-field scale) | MLP + TVA (diagonal covariance) |
> | --- | --- | --- |
> | ETTh1 | 0.438 | 0.439 |
> | ETTh2 | 0.371 | 0.373 |
> | ETTm1 | 0.389 | 0.391 |
> | ETTm2 | 0.278 | 0.279 |
> | Electricity | 0.195 | 0.196 |
> | Solar | 0.234 | 0.224 |
> | Traffic | 0.452 | 0.453 |
> | Weather | 0.260 | 0.261 |
>
> The reported results correspond to the MSE averaged over all prediction lengths $T \in$ {$96, 192, 336, 720$}. As shown in the results, when using a diagonal covariance, we observed slight improvements on the Solar dataset, while yielding comparable performance across the majority of datasets. These results suggest that the commonly used mean-field scale covariance remains effective. We thank the reviewer again for the insightful suggestion and will include these experiments in the Appendix to further strengthen the soundness of the paper.
>
> ---
>
> **References**
>
> [a] Li, Q., Tai, C., and Weinan, E. “Stochastic modified equations and dynamics of stochastic gradient algorithms i: Mathematical foundations.” Journal of Machine Learning Research, 2019.
>
> [b] Li, Z., Malladi, S., and Arora, S. “On the validity of modeling sgd with stochastic differential equations (sdes).” NeurIPS, 2021.
>
> [c] Malladi, S., Lyu, K., Panigrahi, A., and Arora, S. “On the SDEs and scaling rules for adaptive gradient algorithms.” NeurIPS, 2022.

---

> > ### Author Rebuttal · Reviewer_pbzB · 2026-04-03
> >
> > I would like to appreciate the author for the additional experiments. Considering the significance and technical novelty of the paper, I will maintain my score, weak accept.

---

> > > ### Author Response · Authors · 2026-04-05
> > >
> > > We greatly appreciate the reviewer’s time and effort.

---

### Decision · Program_Chairs · 2026-04-30

**Decision:**

Accept (regular)

**Comment:**

Reviewers generally thought the work was sound and well presented, and that the issue addressed is important.  The reviewers also appreciated the simplicity of the method, its compatibility with various architectures and optimizers, and the strong empirical results.  While the reviewers had several questions in common initially about the isotropic scalar approximation to the full gradient-noise covariance and the interaction with other optimizer learning schedules, these were dealt with in the rebuttal.  Two of the reviewers (pbzB and ccPz) asked whether TVA actually decorrelates as advertised; pbzB was convinced by the response (as was I), while ccPz remained slightly skeptical.  There was also some discussion around the strength of the PAC-Bayes theory presented, though this seems fairly reasonable to me (and to many of the reviewers).

Most of the reviewers weakly favored acceptance at the end, and the main concerns expressed had to do with significance and originality.  Yet the technique seems broadly applicable by the assessment of all the reviewers, and the only concrete point I saw in the reviews regarding originality had to do with the connection to orthogonality regularization (which this is not).  In the post-rebuttal discussion, we also had at least one reviewer willing to champion inclusion in the program.  I therefore recommend acceptance.